# Transfer Learning for Latent Variable Network Models

**Akhil Jalan**
Department of Computer Science
UT Austin
akhiljalan@utexas.edu

**Arya Mazumdar**
Halıcıoğlu Data Science Institute & Dept of CSE
UC San Diego
arya@ucsd.edu

**Soumendu Sundar Mukherjee**
Statistics and Mathematics Unit (SMU)
Indian Statistical Institute, Kolkata
ssmukherjee@isical.ac.in

**Purnamrita Sarkar**
Department of Statistics and Data Sciences
UT Austin
purna.sarkar@austin.utexas.edu

## Abstract

We study transfer learning for estimation in latent variable network models. In our setting, the conditional edge probability matrices given the latent variables are represented by $P$ for the source and $Q$ for the target. We wish to estimate $Q$ given two kinds of data: (1) edge data from a subgraph induced by an $o(1)$ fraction of the nodes of $Q$, and (2) edge data from all of $P$. If the source $P$ has no relation to the target $Q$, the estimation error must be $\Omega(1)$. However, we show that if the latent variables are shared, then vanishing error is possible. We give an efficient algorithm that utilizes the ordering of a suitably defined graph distance. Our algorithm achieves $o(1)$ error and does not assume a parametric form on the source or target networks. Next, for the specific case of Stochastic Block Models we prove a minimax lower bound and show that a simple algorithm achieves this rate. Finally, we empirically demonstrate our algorithm's use on real-world and simulated network estimation problems.

## 1 Introduction

Within machine learning and statistics, the paradigm of *transfer learning* describes a setup where data from a source distribution $P$ is exploited to improve estimation of a target distribution $Q$ for which a small amount of data is available. Transfer learning is quite well-studied in learning theory, starting with works such as Ben-David et al. (2006); Cortes et al. (2008); Crammer et al. (2008), and at the same time has found applications in areas such as computer vision (Tzeng et al., 2017a) and speech recognition (Huang et al., 2013). A fairly large body of work in transfer learning considers different types of relations that may exist between $P$ and $Q$, for example, Mansour et al. (2009); Hanneke and Kpotufe (2019, 2022), with emphasis on model selection, multitask learning and domain adaptation. On the other hand, optimal nonparametric rates for transfer learning have very recently been studied, both for regression and classification problems (Cai and Wei, 2021; Cai and Pu, 2024).

In this paper, we study transfer learning in the context of *random network/graph models*. In our setting, we observe Bernoulli samples from the full $n \times n$ edge probability matrix for the source $P$ and only a $n_Q \times n_Q$ submatrix of $Q$ for $n_Q \ll n$. We would like to estimate the full $n \times n$ probability matrix $Q$, using the full source data and limited target data, i.e., we are interested in the task of estimating $Q$ in the partially observed target network, utilizing information from the fully observed source network. This is a natural extension of the transfer learning problem in classification/regression

38th Conference on Neural Information Processing Systems (NeurIPS 2024).

to a network context. However, it is to be noted that network transfer is a genuinely different problem owing to the presence of edge correlations.

While transfer learning in graphs seems to be a fundamental enough problem to warrant attention by itself, we are also motivated by potential applications. For example, metabolic networks model the chemical interactions related to the release and utilization of energy within an organism (Christensen and Nielsen, 2000). Existing algorithms for metabolic network estimation (Sen et al., 2018; Baranwal et al., 2020) and biological network estimation more broadly (Fan et al., 2019; Li et al., 2022) typically assume that some edges are observed for every node in the target network. One exception is Kshirsagar et al. (2013), who leverage side information for host-pathogen protein interaction networks. For the case of metabolic networks, determining interactions *in vivo*[1] requires metabolite balancing and labeling experiments, so only the edges whose endpoints are *both* incident to the experimentally chosen metabolites are observed (Christensen and Nielsen, 2000). For a non-model organism, the experimentally tested metabolites may be a small fraction of all metabolites believed to affect metabolism. However, data for a larger set of metabolites might be available for a model organism.

To study transfer learning on networks, one needs to fix a general enough class of networks that is appropriate for the applications (such as the biological networks mentioned above) and also suitable to capture the transfer phenomenon. The latent variable models defined below appear to be a natural candidate for that.

**Latent Variable Models.** Latent variable network models consist of a large class of models whose edge probabilities are governed by the latent positions of nodes. This includes latent distance models, stochastic block models, random dot product graphs and mixed membership block models (Hoff et al., 2002; Hoff, 2007; Handcock et al., 2007; Holland et al., 1983; Rubin-Delanchy et al., 2022; Airoldi et al., 2008). They can also be unified under graph limits or graphons (Lovász, 2012; Bickel and Chen, 2009), which provide a natural representation of vertex exchangeable graphs (Aldous, 1981; Hoover, 1979). In addition to their theoretical breadth and usefulness, latent variable models are relevant and applicable to real-world settings such as neuroscience Ren et al. (2023), ecology Trifonova et al. (2015), international relations Cao and Ward (2014), political pscyhology Barberá et al. (2015), and education research Sweet et al. (2013).

For unseen latent variables $\boldsymbol{x}_1, \ldots, \boldsymbol{x}_n \in \mathcal{X} \subset \mathbb{R}^d$ and unknown function $f_Q : \mathcal{X} \times \mathcal{X} \to [0,1]$ where $\mathcal{X}$ is a compact set and $d$ an arbitrary fixed dimension, the edge probabilities are:

$$Q_{ij} = f_Q(\boldsymbol{x}_i, \boldsymbol{x}_j). \tag{1}$$

Typically, in network estimation, one observes adjacency matrix $\{A_{ij}\}$ distributed as $\{\text{Bernoulli}(Q_{ij})\}$, and either has to learn $\boldsymbol{x}_i$ or directly estimate $f_Q$. There has been much work in the statistics community on estimating $\boldsymbol{x}_i$ for specific models (usually up to rotation). For stochastic block models, see the excellent survey in Abbe (2017).

Estimating $f_Q$ can be done with some additional assumptions (Chatterjee, 2015). When $f_Q$ has appropriate smoothness properties, one can estimate it by a histogram approximation (Olhede and Wolfe, 2014; Chan and Airoldi, 2014). This setting has also been compared to nonparametric regression with an unknown design (Gao et al., 2015). Methods for network estimation include Universal Singular Value Thresholding (Chatterjee, 2015; Xu, 2018), combinatorial optimization (Gao et al., 2015; Klopp et al., 2017), and neighborhood smoothing (Zhang et al., 2017; Mukherjee and Chakrabarti, 2019).

**Transfer Learning on Networks.** We wish to estimate the target network $Q$. However, we only observe $f_Q$ on $\binom{n_Q}{2}$ pairs of nodes, for a uniformly random subset of variables $S \subset \{1, 2, \ldots, n\}$. We assume $S$ is vanishingly small, so $n_Q := |S| = o(n)$.

Absent additional information, we cannot hope to achieve $o(1)$ mean-squared error. To see this, suppose $f_Q$ is a stochastic block model with 2 communities of equal size. For a node $i \notin S$, no edges incident to $i$ are observed, so its community cannot be learned. Since $n_Q \ll n$, we will attain $\Omega(1)$ error overall. To attain error $o(1)$, we hope to leverage transfer learning from a source $P$ if available. In fact, we give an efficient algorithm to achieve $o(1)$ error, formally stated in Section 2.

**Theorem 1.1** (Theorem 2.3, Informal)**.** *There exists an efficient algorithm such that, if given source data $A_P \in \{0,1\}^{n \times n}$ and target data $A_Q \in \{0,1\}^{n_Q \times n_Q}$ coming from an* appropriate *pair $(f_P, f_Q)$*

---

[1]In the organism, as opposed to *in vitro* (in the lab).

*of latent variable models, outputs $\widehat{Q} \in \mathbb{R}^{n \times n}$ such that*

$$\mathbb{P}\left[\frac{1}{n^2}\|Q - \widehat{Q}\|_F^2 \leq o(1)\right] \geq 1 - o(1).$$

There must be a relationship between P and Q for them to be an *appropriate* pair for transfer learning. We formalize this relationship below.

**Relationship Between Source and Target.** It is natural to consider pairs $(f_P, f_Q)$ such that for all $\boldsymbol{x}, \boldsymbol{y} \in \mathcal{X}$, the difference $(f_P(\boldsymbol{x}, \boldsymbol{y}) - f_Q(\boldsymbol{x}, \boldsymbol{y}))$ is small. For example, Cai and Pu (2024) study transfer learning for nonparametric regression when $f_P - f_Q$ is close to a polynomial in $\boldsymbol{x}, \boldsymbol{y}$. But, requiring $f_P - f_Q$ to be pointwise small does not capture a broad class of pairs in the network setting. For example, if $f_P = \alpha f_Q$. Then $f_P - f_Q = (\alpha - 1)f_Q$ can be far from all polynomials if $f_Q$ is, e.g. a Hölder-smooth graphon.[2] However, under the network model, this means $A_P$ and $A_Q$ are stochastically identical modulo one being $\alpha$ times denser than the other.

We will therefore consider pairs $(f_P, f_Q)$ that are close in some measure of local graph structure. With this in mind, we use a graph distance introduced in Mao et al. (2021) for a different inference problem.

**Definition 1.2** (Graph Distance). *Let $P \in [0,1]^{n \times n}$ be the probability matrix of a graph. For $i, j \in [n], i \neq j$, we define the graph distance between them as follows:*

$$d_P(i,j) := \|(\boldsymbol{e}_i - \boldsymbol{e}_j)^T P^2 (I - \boldsymbol{e}_i \boldsymbol{e}_i^T - \boldsymbol{e}_j \boldsymbol{e}_j^T)\|_2^2,$$

*where $\boldsymbol{e}_i, \boldsymbol{e}_j \in \mathbb{R}^n$ are standard basis vectors.*

Intuitively, this first computes the matrix $P^2$ of common neighbors, and then computes the distance between two rows of the same (ignoring the diagonal elements). We will require that $f_P, f_Q$ satisfy a local similarity condition on the relative rankings of nodes with respect to this graph distance. Since we only estimate the probability matrix of $Q$, the condition is on the latent variables $\boldsymbol{x}_1, \ldots, \boldsymbol{x}_n$ of interest. The hope is that the proximity in graph distance reflects the proximity in latent positions.

**Definition 1.3** (Rankings Assumption at Quantile $h_n$). *Let $(P, Q)$ be a pair of graphs evaluated on $n$ latent positions. We say $(P, Q)$ satisfy the rankings assumption at quantile $h_n \leq 1$ if there exists constant $C > 0$ such that for all $i \in [n]$ and all $j \neq i$, if $j$ belongs to the bottom $h_n$-quantile of $d_P(i, \cdot)$, then $j$ belongs to the bottom $Ch_n$-quantile of $d_Q(i, \cdot)$.*

To further motivate Definition 1.3, recall our motivating example of biological network estimation. Previous works require some form of similarity between networks to enable transfer Sen et al. (2018); Fan et al. (2019); Baranwal et al. (2020). For example, Kshirsagar et al. (2013) require a *commonality hypothesis*: if pathogens A, B target the same neighborhoods in a protein interaction network, one can transfer from A to B. Our rankings assumption similarly posits that to transfer knowledge from A to B, A and B have similar 2-hop neighborhood structures.

Note that Definition 1.3 involves quantiles of graph distances; therefore it is a *relative* condition, because it depends on a rank-ordering within both graphs $P, Q$ before comparison. On the other hand, an *absolute* condition would require that for nodes $i, j \in [n]$, if e.g. $d_P(i, j) < 100$ then $d_Q(i, j) < C \cdot 100$. Our condition is more flexible and will hold for a larger set of graph pairs $(P, Q)$, such as pairs where one graph is much more dense than the other.

Finally, to illustrate Definition 1.3, consider stochastic block models $f_P, f_Q$ with $k_P \geq k_Q$ communities respectively. If nodes $i, j$ are in the same communities then $P\boldsymbol{e}_i = P\boldsymbol{e}_j$, so $d_P(i, j) = 0$. We require that $j$ minimizes $d_Q(i, \cdot)$. This occurs if and only if $d_Q(i, j) = 0$. Hence if $i, j$ belong to the same community in $P$, they are in the same community in $Q$. Note that the converse is not necessary; we could have $Q$ with 1 community and $P$ with arbitrarily many communities.

With the relationship between the source and target defined by the rankings assumption, our contributions are as follows.

**(1) Algorithm for Latent Variable Models.** We provide an efficient Algorithm 1 for latent variable models with Hölder-smooth $f_P, f_Q$. The benefit of this algorithm is that it does not assume a parametric form of $f_P$ and $f_Q$. We prove a guarantee on its error in Theorem 2.3.

---

[2]In fact, Cai and Pu (2024) highlight this exact setting as a direction for future work.

**(2) Minimax Rates.** We prove a minimax lower bound for Stochastic Block Models (SBMs) in Theorem 3.2. Moreover, we provide a simple Algorithm 2 that attains the minimax rate for this class (Proposition 3.4).

**(3) Experimental Results on Real-World Data.** We test both of our algorithms on real-world metabolic networks and dynamic email networks, as well as synthetic data (Section 4).

All proofs are deferred to the Appendix.

## 1.1 Other Related work

Transfer learning has recently drawn a lot of interest both in applied and theoretical communities. The notion of transferring knowledge from one domain with a lot of data to another with less available data has seen applications in epidemiology Apostolopoulos and Bessiana (2020), computer vision Long et al. (2015); Tzeng et al. (2017b); Huh et al. (2016); Donahue et al. (2014); Neyshabur et al. (2020), natural language processing Daumé (2007), etc. For a comprehensive survey see Zhuang et al. (2019); Weiss et al. (2016); Kim et al. (2022). Recently, there have also been advances in the theory of transfer learning Yang et al. (2013); Tripuraneni et al. (2020); Agarwal et al. (2023); Cai and Wei (2021); Cai and Pu (2024); Cody and Beling (2023).

In the context of networks, transfer learning is particularly useful since labeled data is typically hard to obtain. Tang et al. (2016) develop an algorithmic framework to transfer knowledge obtained using available labeled connections from a source network to do link prediction in a target network. Lee et al. (2017) proposes a deep learning framework for graph-structured data that incorporates transfer learning. They transfer geometric information from the source domain to enhance performance on related tasks in a target domain without the need for extensive new data or model training. The SGDA method Qiao et al. (2023) introduce adaptive shift parameters to mitigate domain shifts and propose pseudo-labeling of unlabeled nodes to alleviate label scarcity. Zou et al. (2021) proposes to transfer features from the previous network to the next one in the dynamic community detection problem. Simchowitz et al. (2023) work on combinatorial distribution shift for matrix completion, where only some rows and columns are given. A similar setting is used for link prediction in egocentrically sampled networks in Wu et al. (2018). Zhu et al. (2021) train a graph neural network for transfer based on an ego-graph-based loss function. Learning from observations of the full network and additional information from a game played on the network Leng et al. (2020); Rossi et al. (2022). Wu et al. (2024) study graph transfer learning for node regression in the Gaussian process setting, where the source and target networks are fully observed.

Levin et al. (2022) proposes an inference method from multiple networks all with the same mean but different variances. While our work is related, we do not assume $\mathbb{E}[P_{ij}] = \mathbb{E}[Q_{ij}]$. Cao et al. (2010) do joint link prediction on a collection of networks with the same link function but different parameters.

Another line of related but different work deals with multiplex networks (Lee et al., 2014, 2015; Iacovacci and Bianconi, 2016; Cozzo et al., 2018) and dynamic networks Sarkar and Moore (2005); Kim et al. (2018); Sewell and Chen (2015); Sarkar et al. (2012); Chang et al. (2024); Wang et al. (2023). One can think of transfer learning in clustering as clustering with side information. Prior works consider stochastic block models with noisy label information (Mossel and Xu, 2016; Mazumdar and Saha, 2017a) or oracle access to the latent structure (Mazumdar and Saha, 2017b).

**Notation.** We use lowercase letters $a, b, c$ to denote (real) scalars, boldface $\boldsymbol{x}, \boldsymbol{y}, \boldsymbol{z}$ to denote vectors, and uppercase $A, B, C$ to denote matrices. Let $a \vee b := \max\{a, b\}$ and $a \wedge b := \min\{a, b\}$. For integer $n > 0$, let $[n] := \{1, 2, \ldots, n\}$. For a subset $S \subset [n]$ and $A \in \mathbb{R}^{n \times n}$, let $A[S, S] \in \mathbb{R}^{|S| \times |S|}$ be the principal submatrix with row and column indices in $S$. We denote the $\ell_2$ vector norm as $\|\boldsymbol{x}\| = \|\boldsymbol{x}\|_2$, dot product as $\langle \boldsymbol{x}, \boldsymbol{y} \rangle$, and Frobenius norm as $\|A\| = \|A\|_F$. For functions $f, g : \mathbb{N} \to \mathbb{R}$ we let $f \lesssim g$ denote $f = O(g)$ and $f \gtrsim g$ denote $f = \Omega(g)$. All asymptotics $O(\cdot), o(\cdot), \Omega(\cdot), \omega(\cdot)$ are with respect to $n_Q$ unless specified otherwise.

## 2 Estimating Latent Variable Models with Rankings

In this section, we present a computationally efficient transfer learning algorithm for latent variable models. Algorithm 1 learns the local structure of $P$ based on graph distances (Definition 1.2). For

---

**Algorithm 1** $\widehat{Q}$-Estimation for Latent Variable Models

---

1: **Input:** $A_P \in \{0,1\}^{n \times n}, A_Q \in \{0,1\}^{n_Q \times n_Q}, S \subset [n]$ s.t. $|S| = n_Q$.
2: Initialize $\widehat{Q} \in \mathbb{R}^{n \times n}$ to be all zeroes.
3: For all $i$, all $j \neq i$, compute graph distances:

$$d_{A_P}(i,j) := \|(e_i - e_j)^T (A_P)^2 (I - e_i e_i^T - e_j e_j^T)\|_2^2.$$

4: Fix a bandwidth $h \in (0,1)$ based on $n, n_Q$.
5: **for** $i = 1$ to $n$ **do**
6:      Let $T_i^{A_P}(h) \subset S$ be bottom $h$-quantile of $S$ with respect to $d_{A_P}(i, \cdot)$.
7:      **if** $i \in S$ **then**
8:          Update $T_i^{A_P}(h) \leftarrow T_i^{A_P}(h) \cup \{i\}$.
9:      **end if**
10: **end for**
11: **for** $i = 2$ to $n$ **do**
12:      **for** $1 \leq j < i$ **do**
13:          Compute $\widehat{Q}_{ij} = \widehat{Q}_{ji}$ by averaging:

$$\widehat{Q}_{ij} := \frac{1}{\left|T_i^{A_P}(h)\right|\left|T_j^{A_P}(h)\right|} \sum_{r \in T_i^{A_P}(h)} \sum_{s \in T_j^{A_P}(h)} A_{Q;rs}.$$

14:      **end for**
15: **end for**
16: Return $\widehat{Q}$.

---

each node $i$ of $P$, it ranks the nodes in $S$ with respect to the graph distance $d_P(i, \cdot)$. For most nodes $i, j \in [n]$, none of the edges incident to $i$ or $j$ are observed in $Q$. Therefore, we estimate $\widehat{Q}_{ij}$ by using the edge information about nodes $r, s \in S$ such that $d_P(i, r)$ and $d_P(j, s)$ are small.

Formally, we consider a model as in Eq. (1) with a compact latent space $\mathcal{X} \subset \mathbb{R}^d$ and latent variables sampled i.i.d. from the normalized Lebesgue measure on $\mathcal{X}$. We set $\mathcal{X} = [0,1]^d$ without loss of generality and assume that functions $f : \mathcal{X} \times \mathcal{X} \to [0,1]$ are $\alpha$-Hölder-smooth.

**Definition 2.1.** *Let $f : \mathcal{X} \times \mathcal{X} \to \mathbb{R}$ and $\alpha > 0$. We say $f$ is $\alpha$-Hölder-smooth if there exists $C_\alpha > 0$ such that for all $\boldsymbol{x}, \boldsymbol{x}', \boldsymbol{y} \in \mathcal{X}$,*

$$\sum_{\kappa \in \mathbb{N}^d : \sum_i \kappa_i = \lfloor \alpha \rfloor} \left| \frac{\partial^{\sum_i \kappa_i} f}{\partial x_1^{\kappa_1} \cdots \partial x_d^{\kappa_d}}(\boldsymbol{x}, \boldsymbol{y}) - \frac{\partial^{\sum_i \kappa_i} f}{\partial x_1^{\kappa_1} \cdots \partial x_d^{\kappa_d}}(\boldsymbol{x}', \boldsymbol{y}) \right| \leq C_\alpha \|\boldsymbol{x} - \boldsymbol{x}'\|_2^{\alpha \wedge 1}.$$

To exclude degenerate cases where a node may not have enough neighbors in latent space, we require the following assumption.

**Assumption 2.2** (Assumption 3.2 of Mao et al. (2021))**.** *Let $G$ be a graph on $\boldsymbol{x}_1, \ldots, \boldsymbol{x}_n$. There exist $c_2 > c_1 > 0$ and $\Delta_n = o(1)$ such that for all $\boldsymbol{x}_i, \boldsymbol{x}_j$,*

$$c_1 \|\boldsymbol{x}_i - \boldsymbol{x}_j\|^{\alpha \wedge 1} - \Delta_n \leq \frac{1}{n^3} d_G(i,j) \leq c_2 \|\boldsymbol{x}_i - \boldsymbol{x}_j\|^{\alpha \wedge 1}.$$

The second inequality follows directly from Hölder-smoothness, and the first is shown to hold for e.g. Generalized Random Dot Product Graphs, among others (Mao et al., 2021).

We establish the rate of estimation for Algorithm 1 below.

**Theorem 2.3.** *Let $\widehat{Q}$ be as in Algorithm 1. Let $f_P$ be $\alpha$-Hölder-smooth and $f_Q$ be $\beta$-Hölder-smooth for $\beta \geq \alpha > 0$, and let $c$ be an absolute constant. Suppose $(P, Q)$ satisfy Definition 1.3 at $h_n = c\sqrt{\frac{\log n_Q}{n_Q}}$ and $P$ satisfies Assumption 2.2 with $\Delta_n = O((\frac{\log n}{n_Q})^{\frac{1}{2} \vee \frac{\alpha \wedge 1}{d}})$. Then there exists an absolute constant $C > 0$ such that*

$$\mathbb{P}\left[ \frac{1}{n^2} \|\widehat{Q} - Q\|_F^2 \lesssim \left(\frac{d}{2}\right)^{\frac{\beta \wedge 1}{2}} \left(\frac{\log n}{n_Q}\right)^{\frac{\beta \wedge 1}{2d}} \right] \geq 1 - n_Q^{-C}.$$

To parse Theorem 2.3, consider the effect of various parameter choices. First, observe that our upper bound scales quite slowly with $n$. Even if $n$ is superpolynomial in $n_Q$, e.g. $n = n_Q^{\log n_Q}$, then $\log n = O((\log n_Q)^2) = n_Q^{o(1)}$, so the overall effect on the error is dominated by the $n_Q$ term.

Second, the bound is worse in large dimensions, and scales exponentially in $\frac{1}{d}$. This kind of scaling also occurs in minimax lower bounds for nonparametric regression (Tsybakov, 2009), and upper bounds for smooth graphon estimation (Xu, 2018). However, we caution that nonparametric regression can be quite different from network estimation; it would be very interesting to know the dependence of dimension on minimax lower bounds for network estimation, but to the best of our knowledge this is an open problem. Finally notice that a greater smoothness $\beta$ results in a smaller error, up to $\beta = 1$, exactly as in (Gao et al., 2015; Klopp et al., 2017; Xu, 2018).

## 3   Minimax Rates for Stochastic Block Models

In this section, we will show matching lower and upper bounds for a very structured class of latent variable models, namely, Stochastic Block Models (SBMs).

**Definition 3.1** (SBM). *Let $P \in [0,1]^{n \times n}$. We say $P$ is an $(n,k)$-SBM if there exist $B \in [0,1]^{k \times k}$ and $z : [n] \to [k]$ such that for all $i, j$, $P_{ij} = B_{z(i)z(j)}$. We refer to $z^{-1}(\{j\})$ as community $j \in [k]$.*

We first state a minimax lower bound, proved via Fano's method.

**Theorem 3.2** (Minimax Lower Bound for SBMs). *Let $k_P \geq k_Q \geq 1$ with $k_Q$ dividing $k_P$. Let $\mathcal{F}$ be the family of pairs $(P, Q)$ where $P$ is an $(n, k_P)$-SBM, $Q$ is an $(n, k_Q)$-SBM, and $(P, Q)$ satisfy Definition 1.3 at $h_n = 1/k_P$. Moreover, suppose $S \subset [n]$ is restricted to contain an equal number of nodes from communities $1, 2, \ldots, k_P$ of $P$. Then the minimax rate of estimation is:*

$$\inf_{\widehat{Q} \in [0,1]^{n \times n}} \sup_{(P,Q) \in \mathcal{F}} \mathbb{E}\left[ \frac{1}{n^2} \|\widehat{Q} - Q\|_F^2 \right] \gtrsim \frac{k_Q^2}{n_Q^2}.$$

Note that Definition 1.3 at $h_n = 1/k_P$ implies that the true community structure of $Q$ coarsens that of $P$. The condition that $k_Q$ divides $k_P$ is merely a technical one that we assume for simplicity.

We remark that minimax lower bounds for smooth graphon estimation are established by first showing lower bounds for SBMs, and then constructing a graphon with the same block structure using smooth mollifiers (Gao et al., 2015). Therefore, we expect that Theorem 3.2 can also be extended to the graphon setting, using the same techniques. However, sharp lower bounds for other classes such as Random Dot Product Graphs will likely require different techniques (Xie and Xu, 2020; Yan and Levin, 2023).

**Remark 3.3** (Clustering Regime). *In Appendix A.4 we also prove a minimax lower bound of $\frac{\log k_Q}{n_Q}$ in the regime where the error of recovering the true clustering $z$ dominates. This matches the rate of Gao et al. (2015), but for estimating all $n^2$ entries of $Q$, rather than just the $n_Q^2$ observed entries.*

Theorem 3.2 suggests that a very simple algorithm might achieve the minimax rate. Namely, use both $A_P, A_Q$ to learn communities, and then use only $A_Q$ to learn inter-community edge probabilities. If $(P, Q)$ are in the nonparametric regime where regression error dominates clustering error (called the *weak consistency* or *almost exact recovery* regime), then the overall error will hopefully match the minimax rate.

We formalize this approach in Algorithm 2, and prove that it does achieve the minimax error rate in the weak consistency regime. To this end, we define the signal-to-noise ratio of an SBM with parameter $B \in [0,1]^{k \times k}$ as follows:

$$s := \frac{p - q}{\sqrt{p(1-q)}},$$

where $p = \min_i B_{ii}, q = \max_{i \neq j} B_{ij}$.

**Proposition 3.4** (Error Rate of Algorithm 2). *Suppose $P, Q \in [0,1]^{n \times n}$ are $(n, k_P), (n, k_Q)$-SBMs with minimum community sizes $n_{\min}^{(P)}, n_{\min}^{(Q)}$ respectively. Suppose also that $(P, Q)$ satisfy*

**Algorithm 2** $\widehat{Q}$-Estimation for Stochastic Block Models

1: **Input:** $A_P \in \{0,1\}^{n \times n}, A_Q \in \{0,1\}^{n_Q \times n_Q}, S \subset [n]$ s.t. $|S| = n_Q$.
2: Estimate clusterings $\widehat{Z}_P \in \{0,1\}^{n \times k_P}, \widehat{Z}_Q \in \{0,1\}^{n_Q \times k_Q}$ using Chen et al. (2014) on $A_P, A_Q$ respectively.
3: Let $\widehat{W}_Q \in \mathbb{R}^{k_Q \times k_Q}$ be diagonal with

$$\widehat{W}_{Q;ii} = (\mathbf{1}^T \widehat{Z}_Q \boldsymbol{e}_i)^{-1}.$$

4: Initialize $\widehat{\Pi} \in \{0,1\}^{k_P \times k_Q}$ to be all zeroes.
5: **for** $i \in S$ **do**
6:     Let $j_P \in [k_P], j_Q \in [k_Q]$ be the unique column indices at which row $i$ of $\widehat{Z}_P, \widehat{Z}_Q$ respectively are nonzero.
7:     Let $\widehat{\Pi}_{j_P, j_Q} = 1$.
8: **end for**
9: Let $\widehat{B}_Q \in [0,1]^{k_Q \times k_Q}$ be the block-average:

$$\widehat{B}_Q = \widehat{W}_Q \widehat{Z}_Q^T A_Q \widehat{Z}_Q \widehat{W}_Q.$$

10: **return** $\widehat{Q} := \widehat{Z}_P \widehat{\Pi} \widehat{B}_Q \widehat{\Pi}^T \widehat{Z}_P^T$.

---

*Definition 1.3 at $h_n = n_{\min}^{(P)}/n$. Then if the signal-to-noise ratios are such that: $s_P \geq C(\frac{\sqrt{n}}{n_{\min}^{(P)}} \vee \frac{\log^2(n)}{\sqrt{n_{\min}^{(P)}}})$ and $s_Q \geq C(\frac{\sqrt{n_Q}}{n_{\min(Q)}} \vee \frac{\log^2(n_Q)}{\sqrt{n_{\min}^{(Q)}}})$ for large enough constant $C > 0$, Algorithm 2 returns $\widehat{Q}$ such that*

$$\mathbb{P}\left[\frac{1}{n^2}\|\widehat{Q} - Q\|_F^2 \lesssim \frac{k_Q^2 \log(n_{\min}^{(Q)})}{n_Q^2}\right] \geq 1 - O\left(\frac{1}{n_Q}\right).$$

## 4 Experiments

In this section, we test Algorithm 1 against several classes of simulated and real-world networks. We use quantile cutoff of $h_n = \sqrt{\frac{\log n_Q}{n_Q}}$ for Algorithm 1 in all experiments.

**Baselines.** To the best of our knowledge, our exact transfer formulation has not been considered before in the literature. Therefore, we implement two algorithms as alternatives to Algorithm 1.

*(1) Algorithm 2.* Given $A_P \in \{0,1\}^{n \times n}, A_Q \in \{0,1\}^{n_Q \times n_Q}$, let $k_P = \lceil\sqrt{n}\rceil, k_Q = \lceil\sqrt{n_Q}\rceil$. Compute spectral clusterings $\widehat{Z}_P, \widehat{Z}_Q$ with $k_P, k_Q$ clusters respectively. Let $J_S \in \{0,1\}^{n_Q \times n}$ is such that $J_{S;ij} = 1$ if and only if $i = j$ and $i \in S$. The projection $\widehat{\Pi} \in \mathbb{R}^{k_P \times k_Q}$ solves the least-squares problem $\min_{\Pi \in \mathbb{R}^{k_P \times k_Q}} \|J_S \widehat{Z}_P \Pi - \widehat{Z}_Q\|_F^2$. We compute the $\widehat{\Pi}$ differently from steps 4-7 in Algorithm 2 to account for cases where $Q$ is not a true coarsening of $P$. When $Q$ is a true coarsening of $P$, this reduces to the procedure in steps 4-7. Given $\widehat{Z}_P, \widehat{\Pi}$ we return $\widehat{Q}$ as in Algorithm 2.

*(2) Oracle.* Suppose that an oracle can access data for $Q$ on *all* $n \gg n_Q$ nodes as follows. Fix an error probability $p \in (0,1)$. The oracle is given symmetric $A_Q' \in \{0,1\}^{n \times n}$ with independent entries following a mixture distribution. For all $i, j \in [n]$ with $i < j$ let $X_{ij} \sim \text{Bernoulli}(p)$ and $Y_{ij} \sim \text{Bernoulli}(Q(\boldsymbol{x}_i, \boldsymbol{x}_j))$. Then:

$$A_{Q;ij}' = \mathbb{1}_{i \in S, j \in S} Y_{ij} + (1 - \mathbb{1}_{i \in S, j \in S})((1 - X_{ij})Y_{ij} + X_{ij}(1 - Y_{ij})).$$

Given $A_Q'$, the oracle returns the estimate from Universal Singular Value Thresholding on $A_Q'$ Chatterjee (2015). As $p \to 0$, the error will approach $O(n^{\frac{-2\beta}{2\beta+d}})$ for a $\beta$-smooth network on on $d$-dimensional latent variables (Xu, 2018), so the oracle will outperform any transfer algorithm.

| Source | Target | Alg. 1 | Alg. 2 | Oracle $(p = 0.1)$ | Oracle $(p = 0.3)$ | Oracle $(p = 0.5)$ |
|---|---|---|---|---|---|---|
| Noisy-MMSB $(0.7, 0.3, 0.01)$ | Noisy-MMSB $(0.9, 0.1, 0.01)$ | **0.7473± 0.0648** | 1.3761 ± 1.1586 | *0.9556 ± 0.0633* | 2.2568 ± 0.3107 | 4.2212 ± 0.2825 |
| 0.1-Smooth Graphon | 0.5-Smooth Graphon | *1.7656 ± 0.7494* | 4.5033 ± 1.5613 | **0.5016 ± 0.0562** | 2.4423 ± 0.4574 | 5.7774 ± 0.7126 |
| $\mathbb{R}^{10}$ Latent$(2.5)$ | $\mathbb{R}^{10}$ Latent$(1.0)$ | **0.5744 ± 0.1086** | 1.1773 ± 1.0481 | *0.7715 ± 0.0456* | 2.1822 ± 0.2741 | 4.3335 ± 0.3476 |

Table 1: Comparison of different algorithms on simulated networks. Each cell reports $\widehat{\mu} \pm 2\widehat{\sigma}$ of the mean-squared error over 50 independent trials. Error numbers are all scaled by $1e2$ for ease of reading. Bold: Best algorithm. Emphasis: Second-best algorithm.

**Simulations.** We first test on several classes of simulated networks. For $n_Q = 50, n = 200$, we run 50 independent trials for each setting. We report results for each setting in Table 1, and visualize estimates for stylized examples in Figure 1.

At a glance, Figure 1 shows that Algorithms 1 and 2 both work well on Stochastic Block Models (first row), that only Algorithm 1 works well on graphons (second and third rows), and that the Oracle performs well in all cases.

*Smooth Graphons.* The latent space is $\mathcal{X} = [0, 1]$. We consider graphons of the form $f_\gamma(x, y) = \frac{x^\gamma + y^\gamma}{2}$ where $P, Q$ have different $\gamma$. We denote this the $\gamma$-Smooth Graphon.

*Mixed-Membership Stochastic Block Model.* Set $k_P = \lfloor \sqrt{n} \rfloor, k_Q = \lfloor \sqrt{n_Q} \rfloor$. The latent space $\mathcal{X}$ is the probability simplex $\mathcal{X} = \Delta_{k_P} := \{x \in [0, 1]^{k_P} : \sum_i x_i = 1\} \subset \mathbb{R}^{k_P}$. The latent variables $\boldsymbol{x}_1, \ldots, \boldsymbol{x}_n$ are iid-Dirichlet distributed with equal weights $\frac{1}{k_P}, \ldots, \frac{1}{k_P}$. Then $P_{ij} = \boldsymbol{x}_i^T B_P \boldsymbol{x}_j$ and $Q_{ij} = \Pi(\boldsymbol{x}_i)^T B_Q \Pi(\boldsymbol{x}_j)$, for connectivity matrices $B_P \in [0, 1]^{k_P \times k_P}, B_Q \in [0, 1]^{k_Q \times k_Q}$, and projection $\Pi : \Delta_{k_P} \to \Delta_{k_Q}$ for a fixed subset of $[k_P]$. For parameters $a, b, \epsilon \in [0, 1]$ we generate $B \in [0, 1]^{k \times k}$ by sampling $E \in \text{Uniform}(-\epsilon, \epsilon)^{k \times k}$ and set $B = \text{clip}((a - b)I + b\mathbf{1}\mathbf{1}^T + E, 0, 1)$. We call this Noisy-MMSB$(a, b, \epsilon)$.

*Latent Distance Model.* The latent space is the unit sphere $\mathcal{X} = \mathbb{S}^{d-1} \subset \mathbb{R}^d$. For scale parameter $s > 0$, we call $f_s(\boldsymbol{x}, \boldsymbol{y}) = \exp(-s\|\boldsymbol{x} - \boldsymbol{y}\|_2)$ the $\mathbb{R}^d$-Latent$(s)$ model.

**Discussion.** When the latent dimension is larger than 1 (the Noisy MMSB and Latent Variable Models), our Algorithm 1 is better than both Algorithm 2 and the Oracle with $p = 0.1$. Note that Algorithms 1 and 2 use $\frac{n_Q^2}{n^2} \approx 0.06$ unbiased edge observations from $Q$, while the Oracle with $p = 0.1$ observes $(1 - p)\frac{n^2 - n_Q^2}{n^2} \approx 0.9$ unbiased edge observations in expectation.

**Real-World Data.** Next, we test on two classes of real-world networks. We summarize our dataset characteristics in Table 2. See Appendix C for further details.

Table 2: Dataset Characteristics

| Name | $n$ | Median Degree | Type |
|---|---|---|---|
| BiGG Model iWFL1372 | 251 | 15.00 | Source |
| BiGG Model iPC815 | 251 | 12.00 | Source |
| BiGG Model iJN1463 | 251 | 14.00 | Target |
| EMAIL-EU Days 1-80 | 1005 | 6.92 | Source |
| EMAIL-EU Days 81-160 | 1005 | 7.35 | Target |
| EMAIL-EU Days 561-640 | 1005 | 7.66 | Target |

**Transfer Across Species in Metabolic Networks.** For a fixed organism, a metabolic network has a node for each metabolite, and an edge exists if and only if two metabolites co-occur in a metabolic

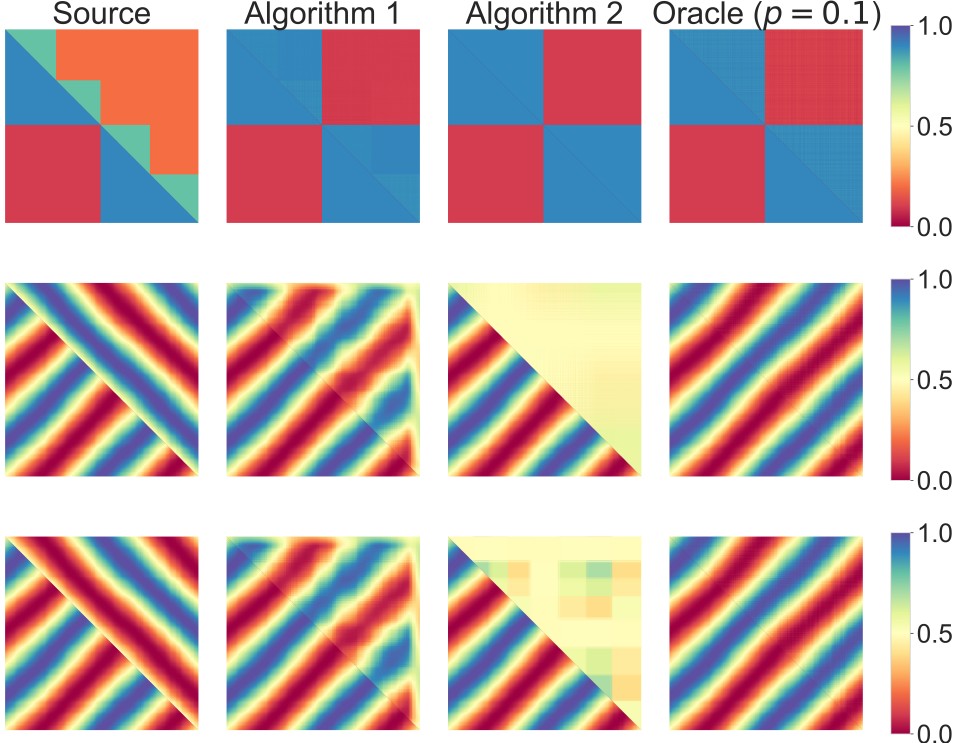

Figure 1: Comparison of algorithms on three source-target pairs ($n = 2000, n_Q = 500$). Each row corresponds to a different source/target pair $(P, Q)$. For a fixed row, the upper triangular part on columns 2, 3, 4 corresponds a $\widehat{Q}$ for a different algorithm. The upper triangular part of column 1 shows the true $P$. The lower triangular part of columns 1, 2, 3, and 4 is identical for a fixed row, and shows the true $Q$. In each heatmap, the lower triangle is the target $Q$. Algorithm 2 performs best when $(P, Q)$ are SBMs (top), while Algorithm 1 is better for smooth graphons (2nd and 3rd rows).

reaction in that organism. We obtain the unweighted metabolic networks for multiple gram-negative bacteria from the BiGG genome-scale metabolic model dataset (King et al., 2016; Norsigian et al., 2020). In the left half of Figure 2, we compare two choices of source organism in estimating the network for BiGG model iJN1463 (*Pseudomonas putida*). For a good choice of source, Algorithm 1 is competitive with the Oracle at $p = 0.1$.

**Transfer Across Time in the Email Interaction Networks.** We use the EMAIL-EU interaction network between $n = 1005$ members of a European research institution across 803 days Leskovec and Krevl (2014); Paranjape et al. (2017). The source graph $A_P$ is the network from day 1 to $\approx 80$ ($[1, 80]$). In Figure 2 we simulate transfer with targets $[81, 160]$ (left) and $[561, 640]$ (right). We visualize results for arbitrary target periods; similar results hold for other targets. Unlike metabolic networks, Algorithm 2 has comparable performance to both our Algorithm 1 and the oracle algorithm with $p \in \{0.01, 0.05\}$. Compared to the metabolic networks, this indicates that the email interaction networks are relatively well-approximated by SBMs, although Algorithm 1 is still the best.

**Additional Experiments and Baseline.** In Appendix B.1, we present additional ablation experiments that test the dependence of Algorithms 1 and 2 on all relevant parameters. We compare their performance to the Oracle baseline with $p = 0.0$ (the non-transfer setting), and an additional baseline adapted from Levin et al. (2022). We find that our Algorithms outperform this new baseline but are worse than the Oracle with $p = 0.0$, as expected. Further, in Appendix B.2, we test our Algorithms and original baselines on a link prediction task in the setting of Figure 2. We find that the relative accuracy of the methods for link prediction is qualitatively similar to that of Figure 2, and the Oracle performs even better with sparsity tuning.

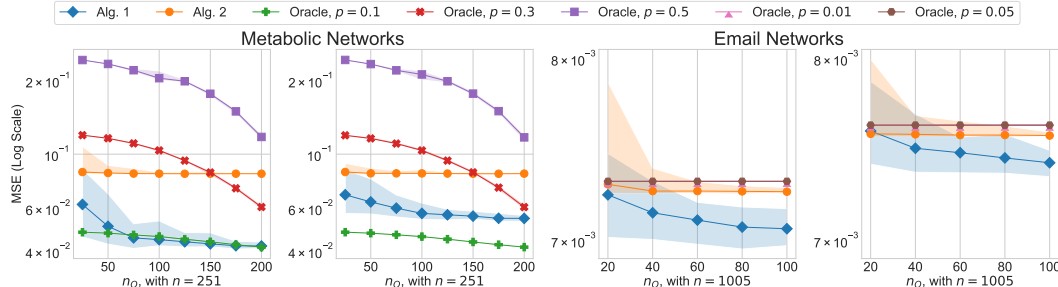

Figure 2: Results of network estimation on real-world data. Shaded regions denote $[1, 99]$ percentile outcomes from 50 trials.
*Left half*: Estimating metabolic network of iJN1463 (*Pseudomonas putida*) with source iWFL1372 (*Escherichia coli W*) leftmost, and source iPC815 (*Yersinia pestis*) second-left.
*Right half*: Using source data from days $1 - 80$ of EMAIL-EU to estimate target days $81 - 160$ (third-left) and target days $561 - 640$ (rightmost). Note that we use smaller values of $p$ for the Oracle in EMAIL-EU.

## 5 Conclusion

In this paper, we study transfer learning for network estimation in latent variable models. We show that there exists an efficient Algorithm 1 that achieves vanishing error even when $n \geq n_Q^{\omega(1)}$, and a simpler Algorithm 2 for SBMs that achieves the minimax rate.

There are several interesting directions for future work.

First, we believe that Algorithm 1 works for moderately sparse networks with population edge density $\Omega(\frac{1}{\sqrt{n}})$. This is because the concentration of empirical graph distance (Algorithm 1 line 3) requires expected edge density $\widetilde{\Omega}(n^{-1/2})$ Mao et al. (2021). It would be interesting to see if a similar approach can work for edge density $\Omega(\frac{\log n}{n})$. For example, in the aforementioned paper it is shown that a variation of the graph distance of Definition 1.2 concentrates at expected edge density $\widetilde{\Omega}(n^{-2/3})$. While is this still far from the $\Omega(\frac{\log n}{n})$ regime, it suggests that variations on the graph distance might ensure our Algorithm 1 works for sparser graphs.

Second, the case of multiple sources is also interesting. We have focused on the case of one source distribution, as in Cai and Wei (2021); Cai and Pu (2024), but expect that our algorithms can be extended to multiple sources as long as they satisfy Definition 1.3.

## Acknowledgments and Disclosure of Funding

We thank the anonymous reviewers for their valuable feedback.

AJ and PS gratefully acknowledge NSF grants 2217069, 2019844, and DMS 2109155.

AM was supported by NSF awards 2217058 and 2133484.

SSM was partially supported by an INSPIRE research grant (DST/INSPIRE/04/2018/002193) from the Dept. of Science and Technology, Govt. of India and a Start-Up Grant from Indian Statistical Institute.

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

# A Proofs

## A.1 Preliminaries

Recall Hoeffding's inequality.

**Lemma A.1** (Hoeffding (1994))**.** *Let $X_1, \ldots, X_n$ be independent random variables such that $a_i \leq X_i \leq b_i$ almost surely for all $i \in [n]$. Then*

$$\mathbb{P}\left[\left|\sum_{i=1}^{n}(X_i - \mathbb{E}[X_i])\right| \geq t\right] \leq 2\exp\left(\frac{-2t^2}{\sum_{i=1}^{n}(b_i - a_i)^2}\right).$$

We also need Bernstein's inequality.

**Lemma A.2** (Bernstein's Inequality)**.** *Let $X_1, \ldots, X_n$ be independent mean-zero random variables with $|X_i| \leq 1$ for all $i$ and $n \geq 5$. Then*

$$\mathbb{P}\left[\left|\frac{1}{n}\sum_{i=1}^{n}X_i\right| \geq t\right] \leq 2\exp\left(\frac{-nt^2}{2(1 + \frac{t}{3})}\right) \leq 2\exp\left(-\frac{nt^2}{4}\right).$$

## A.2 Proof of Theorem 2.3

Throughout this section, let $\mathcal{X} = [0, 1]^d$ and $\mu : \mathcal{X} \to [0, 1]$ be the normalized Lebesgue measure.

We require the following Lemmata.

**Lemma A.3.** *Let $\upsilon \in (0, 1)$ and $\mu : \mathcal{X} \to [0, 1]$ be the normalized Lebesgue measure. Then for all $\boldsymbol{x} \in \mathcal{X}$,*

$$\mu(\mathrm{Ball}(\boldsymbol{x}, 2\upsilon) \cap \mathcal{X}) \geq \mu(\mathrm{Ball}(\boldsymbol{0}, \upsilon) \cap \mathcal{X}).$$

*Proof.* Recall $\mathcal{X} = [0, 1]^d$. Fix $\boldsymbol{x} \in \mathcal{X}, \upsilon > 0$. Note that $\mu(\mathrm{Ball}(\boldsymbol{x}, \upsilon) \cap \mathcal{X})$ is smallest when $\boldsymbol{x}$ is a vertex of the hypercube; therefore take $\boldsymbol{x} \in \{0, 1\}^d$ without loss of generality. Then, note that for each $\boldsymbol{z} \in \mathrm{Ball}(\boldsymbol{x}, \upsilon) \cap \mathcal{X}$, we can find $(2^d - 1)$ other points $\boldsymbol{z}' \in \mathrm{Ball}(\boldsymbol{x}, \upsilon) \setminus \mathcal{X}$ by reflecting subsets of coordinates of $\boldsymbol{z}$ about $\boldsymbol{x}$. There are $2^d - 1$ such nonempty subsets of coordinates. This shows that $\mu(\mathrm{Ball}(\boldsymbol{x}, \upsilon) \cap \mathcal{X}) \geq \mu(\mathrm{Ball}(\boldsymbol{x}, \upsilon))/2^d$ for all $\boldsymbol{x}$. Since $\mu(\mathrm{Ball}(\boldsymbol{x}, \upsilon)) \asymp \upsilon^d$, the conclusion follows. $\square$

We will repeatedly make use of the concentration of latent positions.

**Lemma A.4** (Latent Concentration)**.** *Let $\mathcal{X} = [0, 1]^d$ and $\mu$ denote the normalized Lebesgue measure on $\mathcal{X}$. Suppose $\boldsymbol{x}_1, \ldots, \boldsymbol{x}_n \sim \mathcal{X}$ are sampled iid and uniformly at random from $\mu$. Fix some $T \subset \mathcal{X}$ such that $\mu(T) = \upsilon$. Then*

$$\mathbb{P}\left[\left|\upsilon n - |\{j \in [n] : \boldsymbol{x}_j \in T\}|\right| \geq 10\sqrt{\frac{\log n}{n}}\right] \leq n^{-10}.$$

*Proof.* Let $X_i$ be an indicator variable that equals 1 if $\boldsymbol{x}_i \in T$ and zero otherwise. Notice the $X_i$ are iid and bounded within $[0, 1]$. Moreover, $\sum_i \mathbb{E}[X_i] = n\mu(T)$. Therefore by Hoeffding's inequality, for any $t > 0$,

$$\mathbb{P}[|\upsilon n - |\{j \in [n] : \boldsymbol{x}_j \in T\}|| \geq t] \leq 2\exp\left(\frac{-2t^2}{n}\right).$$

Setting $t = 10\sqrt{\frac{\log n}{n}}$ gives the result. $\square$

**Corollary A.5.** *Let $\epsilon > 0$. For $i \in [n]$ let $\epsilon_i' > 0$ be $\epsilon_i' := \sup\{v > 0 : \mu(\text{Ball}(\boldsymbol{x}_i, v) \cap \mathcal{X}) \leq \epsilon$. Let $T_i := \text{Ball}(\boldsymbol{x}_i, \epsilon_i') \cap \mathcal{X}$. Let $u_i(S) := |\{j \in S : \boldsymbol{x}_j \in T_i\}|$ denote the number of members of $S$ landing in $T_i$. Then*

$$\mathbb{P}\left[\forall i \in [n] : |u_i(S) - n_Q \epsilon| \leq 10\sqrt{\frac{\log n}{n_Q}}\right] \geq 1 - n^{-8}.$$

*Proof.* Notice that each $T_i$ has Lebesgue measure $\epsilon$ by definition. Therefore $\mathbb{E}[u_i(S)] = n_Q \epsilon$. Since $S$ has $n_Q$ members, setting $t = 10\sqrt{\frac{\log n}{n_Q}}$ in the statement of Lemma A.4 and taking a union bound over all $i \in [n]$ gives the conclusion. $\square$

We will decompose the error of Algorithm 1 into two parts.

**Proposition A.6.** *Let $\widehat{Q} \in [0,1]^{n \times n}$ be the estimator from Algorithm 1. Then*

$$\frac{1}{n^2}\|Q - \widehat{Q}\|_F^2 \leq \frac{2}{n^2}\sum_{i,j \in [n]}(J_S(i,j) + J_B(i,j)),$$

*where $J_S, J_B$ are the smoothing and Bernoulli errors respectively:*

$$J_S(i,j) := \frac{1}{|T_i|^2\,|T_j|^2}\left(\sum_{r \in T_i, s \in T_j} Q_{ij} - Q_{rs}\right)^2;$$

$$J_B(i,j) := \frac{1}{|T_i|^2\,|T_j|^2}\left(\sum_{r \in T_i, s \in T_j} Q_{rs} - A_{Q;rs}\right)^2.$$

Controlling the Bernoulli errors is relatively straightforward.

**Proposition A.7.** *Let $h$ be the bandwidth of Algorithm 1. The Bernoulli error is at most $O(\frac{\log n}{m})$ with probability $\geq 1 - n^{-8}$, where $m = h^2 n_Q^2$.*

*Proof.* Fix $i, j \in [n]$. We will bound the maximum Bernoulli error $J_S(i,j)$ over $i, j$, which suffices to bound the average. Let $m = |T_i|\,|T_j|$. We want to bound:

$$\left|\frac{1}{|T_i|\,|T_j|}\sum_{r \in T_i, s \in T_j}(Q_{rs} - A_{Q;rs})\right|^2.$$

Notice each summand is bounded within $\pm\frac{1}{m}$. Bernstein's inequality gives:

$$\mathbb{P}\left[\left(\frac{1}{|T_i|\,|T_j|}\sum_{r \in T_i, s \in T_j} Q_{rs} - A_{Q;rs}\right)^2 \geq t^2\right] \leq 2\exp(-0.5t^2 m).$$

Setting $t = C\sqrt{\frac{\log n}{m}}$ for large enough $C = O(1)$, a union bound tells us that with probability $\geq 1 - n^{-8}$, the Bernoulli error is bounded by $t^2$. $\square$

**Corollary A.8.** *The Bernoulli error is at most $O(\sqrt{\frac{\log n_Q}{n_Q}})$ with probability $\geq 1 - n_Q^{-4}$.*

The rest of this section is devoted to bounded the smoothing errors $J_S(i,j)$.

### A.2.1 Latent Distance to Graph Distance

We claim that if nodes are close in the latent space then they are close in graph distance.

**Proposition A.9.** *Suppose that $\|\boldsymbol{x}_i - \boldsymbol{x}_r\| \leq \epsilon$ and $Q$ is $\beta$-smooth. Then $d_Q(i, r) \leq C_\beta^2 n^3 \epsilon^{2(\beta \wedge 1)}$.*

*Proof.* We the use smoothness of $Q$. By definition there exists $C_\beta > 0$ such that $Q_{ki} - Q_{kr} \leq C_\beta \|\boldsymbol{x}_i - \boldsymbol{x}_r\|^{\beta \wedge 1}$. Therefore,

$$
\begin{aligned}
d_Q(i, r) &= \sum_{\ell \neq i, r} \left| (Q^2)_{\ell i} - (Q^2)_{\ell r} \right|^2 \\
&= \sum_{\ell \neq i, r} \left( \sum_{k \in [n]} Q_{\ell k}(Q_{ki} - Q_{kr}) \right)^2 \\
&\leq \sum_{\ell \neq i, r} \sum_{k \in [n]} Q_{\ell k}^2 C_\beta^2 \epsilon^{2(\beta \wedge 1)} \\
&\leq n^3 C_\beta^2 \epsilon^{2(\beta \wedge 1)}.
\end{aligned}
$$
$\square$

We can now bound the minimum sizes of the neighborhoods using the concentration of latent positions and the smoothness of the graphon.

**Lemma A.10** (Vershynin (2018)). *The volume of a ball of radius $r > 0$ in $\mathbb{R}^d$ is $\frac{\sqrt{\pi}^d}{\Gamma(d/2+1)} r^d$, where $\Gamma(\cdot)$ is the $\Gamma$ function.*

**Proposition A.11.** *Let $C_d = (\Gamma(\frac{d}{2} + 1))^{1/d}$. Let $C_0, C'$ be constants. If $\upsilon_n \geq C \cdot C_d (\sqrt{\frac{\log n}{n_Q}})^{1/d}$ for large enough constant $C > 0$, and $g_n = C_0 C_\beta^2 n^2 (\upsilon_n)^{2(\beta \wedge 1)}$, then with probability $\geq 1 - n^{-6}$ for all $i \in [n]$ the neighborhood size is $|\{r : d_Q(i, r) \leq g_n\}| \geq C' n_Q \sqrt{\frac{\log n}{n_Q}}$.*

*Proof.* Fix $i \in [n]$ and $\upsilon_n > 0$. Let $\epsilon_i$ denote the Lebesgue measure of $\mathrm{Ball}(\boldsymbol{x}_i, \upsilon_n) \cap \mathcal{X}$. By Lemma A.3 and Lemma A.10, for all $i$, $\epsilon_i \geq (\frac{\sqrt{\pi} \upsilon_n}{2 C_d})^d = (\frac{0.5 \sqrt{\pi} \upsilon_n}{C_d})^d$. Let $\epsilon = \min_{i \in [n]} \epsilon_i$.

By Corollary A.5, with probability $\geq 1 - n^{-8}$, there are $n_Q \epsilon - C \sqrt{\frac{\log n}{n_Q}}$ members $j$ of $S$ such that $\|\boldsymbol{x}_i - \boldsymbol{x}_j\| \leq \upsilon_n$. A union bound over $i$ gives the result simultaneously for all $i$ with probability $\geq 1 - n^{-6}$.

From Proposition A.9, it follows that for all $i \in [n]$,

$$
\left| \{ r \in S : d_Q(i, r) \leq C_\beta^2 n^2 (2\upsilon_n')^{2(\beta \wedge 1)} \} \right| \geq n_Q \epsilon - 10 \sqrt{\frac{\log n}{n_Q}}.
$$

Choosing $\upsilon_n \geq C \cdot C_d (\frac{\log n}{n_Q})^{\frac{1}{2d}}$ for large enough $C > 0$ gives the conclusion. $\square$

### A.2.2 Graph Distance Concentration

Next, we show that the empirical graph distance concentrates to the population distance.

**Proposition A.12.** *For any arbitrary symmetric $P \in [0, 1]^{n \times n}$, we have, for all $i, j$ simultaneously with probability at least $\geq 1 - O(n^{-8})$, that*

$$
|d_{A_P}(i, j) - d_P(i, j)| \leq O(n^2 \log n) + O(n^{2.5} \sqrt{\log n}).
$$

*Proof.* Fix $i, j$. Let $C_{ij} := (A_P^2)_{ij}$. By Mao et al. (2021) A.1, we have $C_{ij} = (P^2)_{ij} + t_{ij}$ for an error term $t_{ij}$ such that $\mathbb{P}[\forall i, j : |t_{ij}| \leq 10\sqrt{n \log n}] \geq 1 - n^{-10}$. Then,

$$
\begin{aligned}
|d_{A_P}(i,j) - d_P(i,j)| &= \left| \sum_{\ell \neq i,j} \left( (C_{i\ell} - C_{j\ell})^2 - ((P^2)_{i\ell} - (P^2)_{j\ell})^2 \right) \right| \\
&= \sum_{\ell \neq i,j} \left| (t_{i\ell} + t_{j\ell})^2 + 2(t_{i\ell} + t_{j\ell})((P^2)_{i\ell} - (P^2)_{j\ell}) \right| \\
&\leq O(n^2 \log n) + O\left( \sqrt{n \log n} \sum_{\ell \neq i,j} ((P^2)_{i\ell} - (P^2)_{j\ell}) \right).
\end{aligned}
$$

Finally, notice that all entries of $P^2$ are of size $O(n)$, so the conclusion follows. $\qquad\square$

Finally, we will show that taking the restriction of the graph distance $T_i^P$ to nodes in $S \subset [n]$ does not incur too much error.

**Proposition A.13.** *Suppose $n = n_Q^{O(1)}$. Then there exists a constant $C$ such that if $h_0 \geq C\sqrt{\frac{\log n}{n_Q}} + \Delta_n$, then for all $i, r$ simultaneously, $r \in T_i^{A_P}(h_0)$ implies $r \in T_i^P(h_2)$ for some $h_2 = O(h)$ with probability $\geq 1 - O(n^{-5})$.*

*Proof.* Let us introduce the notation $T_i^{P,S}(h)$ to denote the bottom $h$-quantile of $\{d_P(i,j) : j \in S\}$. In this notation, $T_i^{A_P}(h) := T_i^{A_P,S}(h)$ since we restrict the quantile to nodes in $S$. From Proposition A.12 and Assumption 2.2, we know that if $n \geq n_Q$ then for $h_0 \leq h_1 - 20\sqrt{\frac{\log n}{n}} - \Delta_n$ we have $T_i^{A_P}(h_0) \subseteq T_i^{P,S}(h_1)$ simultaneously for all $i \in [n]$ with probability $\geq 1 - O(n^{-8})$. It remains to compare $T_i^{P,S}(h_1)$ with $T_i^P(h_2)$ for some $h_2$.

We claim that if $h_2 \geq 30\sqrt{\frac{\log n_Q}{n_Q}}$ then $\mathbb{P}[\forall i \, |T_i^P \cap S| \geq h_2 n_Q - 3\sqrt{n_Q \log n_Q}] \geq 1 - O(n_Q^{-2})$. To see this, fix $i \in [n]$ and consider $T_i^P(h_2)$. For $j \in S$, let $X_j$ be the indicator variable:

$$
X_j = \begin{cases} 1 & \text{if } j \in T_i^P(h_2), \\ 0 & \text{otherwise.} \end{cases}
$$

Notice that $|T_i^P(h_2) \cap S| = \sum_{j \in S} X_j$. By Hoeffding's inequality, since $\mathbb{E}[\sum_{j \in S} X_j] = h_2 n_Q$ and $|X_j - h_2| \leq 1$ for all $j$, we have

$$
\mathbb{P}\left[ \left| |T_i^P(h_2) \cap S| - h_2 n_Q \right| \geq 3\sqrt{n_Q \log n} \right] \leq 2 \exp\left( -\frac{6 n_Q^2 \log n}{n_Q^2} \right) \leq 2n^{-6}.
$$

Taking a union bound over all $i \in [n]$ shows the claim holds with probability $\geq 1 - O(n^{-5})$. Therefore we set $h_1 \leq h_2 - 3.1\sqrt{\frac{\log n}{n_Q}}$ then $j \in T_i^{P,S}(h_1)$ implies $j \in T_i^P(h_2)$.

The conclusion follows with $C = 24\sqrt{\frac{\log n}{\log n_Q}} = O(1)$. $\qquad\square$

The ranking condition (Definition 1.3) then allows us to translate between graph distances in $A_P$ and $Q$.

**Corollary A.14.** *Suppose that Definition 1.3 holds for $(P, Q)$ at $h_n = c\sqrt{\frac{\log n_Q}{n_Q}} + \Delta_n$, for large enough constant $c > 0$. Suppose $n_Q \leq n \leq n_Q^{O(1)}$. Then for $h > h_n$ and $r \in T_i^{A_P}(h)$, it follows that $r \in T_i^Q(h_3)$ for some $h_3 = O(h)$. The statement holds simultaneously for all $i, r$ with probability $\geq 1 - O(n^{-5})$.*

### A.2.3 Control of Smoothing Error

We will decompose smoothing error into a sum of two terms called $E_{S,1}$ and $E_{S,2}$. The control of $E_{S,1}$ is relatively straightforward.

**Lemma A.15.** *The total smoothing error can be bounded with two terms:*

$$\frac{2}{n^2} \sum_{i,j\in[n]} J_S(i,j) \leq E_{S,1} + E_{S,2},$$

*where*

$$E_{S,1} := \frac{C}{n} \max_{j\in[n],s\in T_j} \|Q(e_j - e_s)\|_2^2;$$

$$E_{S,2} := \frac{4}{n^2} \sum_{i\in[n]} \frac{1}{|T_i|} \mathbb{E}\left[\sum_{r\in T_i} \sum_{j\in[n]} \sum_{s\in T_j} (Q_{rj} - Q_{rs})^2.\right]$$

*Proof.* Note that

$$\frac{2}{n^2} \sum_{i,j\in[n]} J_S(i,j) = \frac{2}{n^2} \sum_{i,j\in[n]} \frac{1}{|T_i|^2 |T_j|^2} \mathbb{E}\left[\left(\sum_{r\in T_i, s\in T_j} Q_{ij} - Q_{rs}\right)^2\right]$$

$$\leq \frac{2}{n} \sum_{i\in[n]} \frac{1}{n|T_i|} \sum_{j\in[n]} \frac{2}{|T_j|} \mathbb{E}\left[\sum_{r\in T_i, s\in T_j} (Q_{ij} - Q_{rj})^2 + (Q_{rj} - Q_{rs})^2\right]$$

$$= \frac{4}{n} \sum_{i\in[n]} \frac{1}{n|T_i|} \mathbb{E}\left[\sum_j \frac{1}{|T_j|} \left(\sum_{r\in T_i} (Q_{ij} - Q_{rj})^2 + \sum_{r\in T_i} \sum_{s\in T_j} (Q_{rj} - Q_{rs})^2\right)\right].$$

The second inner summand is precise $E_{S,2}$. For $E_{S,1}$, notice that $|T_i| = |T_j| = h(n_Q - 1)$ by definition. Therefore

$$\sum_j \frac{1}{|T_j|} \sum_{r\in T_i} (Q_{ij} - Q_{rj})^2 = \frac{1}{h(n_Q - 1)} \sum_{r\in T_i} \sum_j (Q_{ij} - Q_{rj})^2 \leq 2 \max_{r\in T_i} \|(e_i - e_r)^T Q\|_2^2. \quad \square$$

We can now bound $E_{S,1}$ in terms of graph distances.

**Lemma A.16.** *The smoothing error term $E_{S,1}$ can be bounded as follows:*

$$E_{S,1} \leq \frac{2}{n} \max_{i\in[n],r\in T_i} \sqrt{d_Q(i,r)} + \frac{2c}{\sqrt{n}}$$

*for some constant $c > 0$.*

*Proof.* Fix $i \in [n]$ and $r \in T_i$. We have

$$\|Q(e_i - e_r)\|_2^2 \leq \|e_i - e_r\|_2 \|Q^T Q(e_i - e_r)\|_2$$
$$\leq 2\|Q^2(e_i - e_r)\|_2.$$

Now we will pass to graph distances. Let $e_{ab} := ((Q^2)_{aa} - (Q^2)_{ab})^2$ for $a, b \in [n]$. Notice that $\|Q^2(e_i - e_r)\|_2 = \sqrt{d_Q(i,r) + e_{ir} + e_{ri}}$. Moreover, $\sqrt{e_{ir} + e_{ri}} \leq 2\sqrt{n}$ since the entries of $Q^2$ are individually bounded by $O(n)$. The conclusion follows. $\quad \square$

**Proposition A.17.** *Suppose $\Delta_n = O(\sqrt{\frac{\log n}{n_Q}})$. Let $C_d$ be the constant of Proposition A.11. Then if the bandwidth of Algorithm 1 is $h_n = C\sqrt{\frac{\log n}{n_Q}}$, for a constant $C = O(1)$, then the smoothing error $E_{S,1}$ is at most*

$$E_{S,1} \leq C_2 C_d^{\beta \wedge 1} \left(\sqrt{\frac{\log n_Q}{n_Q}}\right)^{\frac{\beta \wedge 1}{d}}$$

*for some $C_2 = O(1)$, with probability $\geq 1 - O(n^{-6})$.*

*Proof.* Fix $i \in [n]$ and $r \in T_i^{A_P}(h_n)$. By Corollary A.14, if $h_n \geq C\sqrt{\frac{\log n}{n_Q}} + \Delta_n$ for a large enough constant $C > 0$, then there exists constant $C_2 > 0$ such that the following holds. With probability $\geq 1 - O(n^{-5})$, for all $i \in [n]$ and $r \in S$, $r \in T_i^Q(C_2 h_n)$,

Let $v_n = CC_d(\sqrt{\frac{\log n}{n_Q}})^{1/d}$ for $C_d$ as in Proposition A.11 and $C > 0$ large enough constant. Then by Proposition A.11 the set of $s \in S$ such that $d_Q(i,r) \leq C_0 C_\beta^2 n^2 (v_n)^{2(\beta \wedge 1)}$ has size at least $C_2 n_Q \sqrt{\frac{\log n}{n_Q}}$. The statement holds for all $i$ simultaneously with probability at least $1 - O(n^{-6})$. Therefore for all $i \in [n]$ and $r \in T_i^{A_P}(h_n)$, we have

$$d_Q(i,r) \leq C_0 C_\beta^2 n^2 (v_n)^{2(\beta \wedge 1)}$$

for some $C_0, C_\beta = O(1)$, with probability $\geq 1 - O(n^{-6})$. By Lemma A.16 we conclude that $E_{S,1}$ is bounded by $2v_n^{\beta \wedge 1} + \frac{2}{\sqrt{n}}$ with the same probability. $\square$

### A.2.4 Control of the Second Smoothing Error

In this section, we show that the second smoothing error can be controlled in terms of $E_{S,1}$. We will need to track the following quantity.

**Definition A.18** (Membership Count). *For $r \in S$ and bandwidth $h$, distance cutoff $\epsilon$, the $P$-neighborhood count of $r$ is $\psi_P(r) := \left|\{j \in [n] : r \in T_j^P(h, \epsilon)\}\right|$.*

In words, $\psi_P(r)$ counts the number of nodes $j \in [n]$ such that $r$ lands in the neighborhood of $j$ in our algorithm. While we know that $\left|T_j^P(h)\right| \leq h n_Q$ always, simply applying the pigeonhole principle gives too weak of a bound on membership counts. The base case is that there may be a "hub" node $r$ lands in $T_j^P(h)$ for all $j$. We will show that there can be no such hub node.

Supposing that we can control of the empirical count $\psi_{A_P}$, we show that the smoothing error can be bounded.

**Proposition A.19.** *Let $h_n$ be the bandwidth. Then*

$$E_{S,2} \leq O\left(\frac{E_{S,1}}{h_n n}\right) \cdot \max_{r \in [n]}(\psi_{A_P}(r)).$$

*Proof.* Rearranging terms, we have

$$E_{S,2} = \frac{1}{n^2 h^2 n_Q^2} \sum_{i,j \in [n], r \in T_i, s \in T_j} (Q_{rj} - Q_{rs})^2$$

$$= \frac{1}{n^2 h^2 n_Q^2} \sum_{r \in S} \psi_{A_P}(r) \sum_{j,s} (Q_{rj} - Q_{rs})^2$$

$$= \frac{n_Q}{n^2 h^2 n_Q^2} \mathop{\mathbb{E}}_{r \in S} \left[\psi_{A_P}(r) \sum_{j,s} (Q_{rj} - Q_{rs})^2\right]$$

$$= \frac{n_Q}{n^2 h^2 n_Q^2} \mathop{\mathbb{E}}_{r \in [n]} \left[\psi_{A_P}(r) \sum_{j,s} (Q_{rj} - Q_{rs})^2\right],$$

where the last step follows because $j, s$ do not depend on $i, r$ and because $S \subset [n]$ is chosen uniformly at random. Now, we will control the expectation by passing to a row sum, which is handled by $E_{S,1}$.

$$\mathop{\mathbb{E}}_{r \in [n]} \left[\psi_{A_P}(r) \sum_{j,s} (Q_{rj} - Q_{rs})^2\right] \leq \max_{r \in [n]} \left(\frac{\psi_{A_P}(r)}{n}\right) \cdot \sum_{j \in [n]} \sum_{s \in T_j} \|Q(e_j - e_s)\|_2^2.$$

Recall that $n^2 n_Q h_n E_{S,1} = \Omega\left(\sum_{j \in [n]} \sum_{s \in T_j} \|Q(e_j - e_s)\|_2^2\right)$. Hence we conclude that

$$E_{S,2} \leq O\left(\frac{E_{S,1}}{h_n n}\right) \cdot \max_{r \in [n]}(\psi_{A_P}(r)). \qquad \square$$

We therefore must show that $\max_{r \in S} \psi_{A_P}(r) \leq O(hn)$ with high probability.

**Proposition A.20** (Population Version). *Suppose Assumption 2.2 holds for $P$ with $c_1 < c_2$ and $\Delta_n = O((\frac{\log n}{n_Q})^{\frac{1}{2} \vee \frac{\alpha \wedge 1}{d}})$. Then if $h \leq C\sqrt{\frac{\log n}{n_Q}}$ for large enough constant $C > 0$, then we have $\max_{r \in S} \psi_P(r) \leq O(hn)$ with probability at least $1 - O(n_Q^{-8})$.*

*Proof.* Fix $r \in S$. Let $C_d$ be as in Proposition A.11. Suppose that $\epsilon = C_d(C + 10)\sqrt{\frac{\log n_Q}{n_Q}}^{-1/d}$ and $h = C\sqrt{\frac{\log n_Q}{n_Q}}$. Now, we will claim that for large enough constant $c > 0$, that $\psi_P(r)$ is at most the size of $\text{Ball}(\boldsymbol{x}_r, c\epsilon) \cap \{\boldsymbol{x}_1, \ldots, \boldsymbol{x}_n\}$.

Suppose that $c > 0$ is a large enough constant. Now suppose that $\boldsymbol{x}_j$ is such that $\|\boldsymbol{x}_j - \boldsymbol{x}_r\| \geq c\epsilon$. We can lower bound the graph distance using Assumption 2.2, as:

$$d_P(r, j) := \|(e_r - e_j)^T P^2(I - e_r e_r^T - e_j e_j^T)\|_2^2 \geq c_1 n^3 (c\epsilon)^{2(\alpha \wedge 1)} - n^3 \Delta_n.$$

On the other hand, suppose that $i \in S$ is such that $\|\boldsymbol{x}_i - \boldsymbol{x}_j\| \leq \epsilon$. Then $d_P(i, j) \leq C_\alpha^2 n^3 \epsilon^{2(\alpha \wedge 1)}$ by Proposition A.9. Therefore since $\epsilon = C_d(C + 10)\sqrt{\frac{\log n_Q}{n_Q}}^{-1/d}$ and $\Delta_n = O((\frac{\log n}{n_Q})^{\frac{1}{2} \vee \frac{\alpha \wedge 1}{d}})$, for large enough $c_1 > 0$ we have

$$d_P(r, j) := \|(e_r - e_j)^T P^2(I - e_r e_r^T - e_j e_j^T)\|_2^2 \geq \frac{c_1}{2} n^3 (c\epsilon)^{2(\alpha \wedge 1)}.$$

Then, if we choose $c > 0$ such that $c^{2(\alpha \wedge 1)} > \frac{2C_\alpha^2}{c_1}$, then $d_P(i, j) < d_P(r, j)$.

Next, from our choices of $h, \epsilon$, by Corollary A.5, simultaneously for all $i \in [n]$ there are at least $hn_Q$ nodes in $S$ that have distance $\leq \epsilon$ in latent space from $\boldsymbol{x}_i$, with probablity $\geq 1 - O(n_Q^{-6})$.

Therefore, if $\boldsymbol{x}_r \notin \text{Ball}(\boldsymbol{x}_j, c\epsilon) \cap \{\boldsymbol{x}_1, \ldots, \boldsymbol{x}_n\}$ then $r \notin T_j^P(h)$. This implies that $\psi_P(r) \leq |\{\text{Ball}(\boldsymbol{x}_r, 2c\epsilon) \cap \{\boldsymbol{x}_1, \ldots, \boldsymbol{x}_n\}\}|$. We can bound the size of this ball with Lemma A.4. Notice the Lebesgue measure of $\text{Ball}(\boldsymbol{x}_r, 2c\epsilon) \cap [0, 1]$ is at most $(\frac{4c\epsilon}{C_d})^d$. Therefore, since $\boldsymbol{x}_i$ are chosen iid from the Lebesgue measure on $\mathcal{X}$, with probability at least $\geq 1 - O(n_Q^{-10})$, we have

$$\frac{1}{n}|\text{Ball}(\boldsymbol{x}_r, 2c\epsilon) \cap \{\boldsymbol{x}_1, \ldots, \boldsymbol{x}_n\}| \leq 2c\epsilon + 10\sqrt{\frac{\log n}{n}}.$$

The right-hand side is bounded by $O(h)$ if $n \geq n_Q$. Taking a union bound over all $r \in S$ gives the conclusion. $\square$

We conclude with the desired upper bound.

**Proposition A.21** (Bound on $\psi_{A_P}(r)$). *Suppose Assumption 2.2 holds for $P$ with $c_1 < c_2$ and $\Delta_n = O((\frac{\log n}{n_Q})^{\frac{1}{2} \vee \frac{\alpha \wedge 1}{d}})$. Then if $h \leq C_0\sqrt{\frac{\log n_Q}{n_Q}}$ for small enough constant $C_0$, then we have $\max_{r \in S} \psi_{A_P}(r) \leq O(hn)$ with probability at least $1 - O(n_Q^{-8})$.*

*Proof.* By Proposition A.12, with probability at least $1 - O(n_Q^{-8})$, we have for all $r \in S, j \in [n]$ simultaneously that

$$d_{A_P}(r, j) \geq d_P(r, j) - O(n^{2.5}\sqrt{\log n})$$
$$\geq (1 - O(\frac{1}{\sqrt{n}}))d_P(r, j).$$

Similarly, $d_{A_P}(r, j) \leq (1 + O(\frac{1}{\sqrt{n}}))d_P(r, j)$. We conclude that $\psi_{A_P}(r) \leq 2\psi_P(r) = O(hn)$ with probability $\geq 1 - O(n_Q^{-8})$. $\square$

### A.2.5 Overall Error

We can bound $C_d := \Gamma(\frac{d}{2} + 1)^{1/d}$ with the elementary inequality.

**Lemma A.22.** *Let $C_d := \Gamma(\frac{d}{2} + 1)^{1/d}$. Then $C_d \leq \sqrt{d/2}$.*

*Proof of Theorem 2.3.* By Proposition A.21 and Prop A.19, we have that $E_{S,1} \leq O(E_{S,1})$ with probability at least $1 - O(n_Q^{-8})$. Therefore by Proposition A.17,

$$\mathbb{P}\left[E_{S,1} + E_{S,2} \leq O\left(C_d^{\beta \wedge 1}\left(\frac{\log n}{n_Q}\right)^{\frac{\beta \wedge 1}{2d}}\right)\right] \geq 1 - O(n_Q^{-6}).$$

By Lemma A.22, $C_d \leq \sqrt{d/2}$. Finally, by Corollary A.8, the Bernoulli error is bounded by $O(\sqrt{\frac{\log n_Q}{n_Q}})$ with probability $\geq 1 - O(n_Q^{-4})$. Applying a union bound over the two kinds of error and Lemma A.15 gives the result. $\qquad\square$

### A.3 Proof of Theorem 3.2

Recall the Gilbert-Varshamov code (Guruswami et al., 2019).

**Theorem A.23** (Gilbert-Varshamov)**.** *Let $q \geq 2$ be a prime power. For $0 < \epsilon < \frac{q-1}{q}$ there exists an $\epsilon$-balanced code $C \subset \mathbb{F}_q^n$ with rate $\Omega(\epsilon^2 n)$.*

We will use the following version of Fano's inequality.

**Theorem A.24** (Generalized Fano Method, Yu (1997))**.** *Let $\mathcal{P}$ be a family of probability measures, $(\mathcal{D}, d)$ a pseudo-metric space, and $\theta : \mathcal{P} \to \mathcal{D}$ a map that extracts the parameters of interest. For a distinguished $P \in \mathcal{P}$, let $X \sim P$ be the data and $\widehat{\theta} := \widehat{\theta}(X)$ be an estimator for $\theta(P)$.*

*Let $r \geq 2$ and $\mathcal{P}_r \subset \mathcal{P}$ be a finite hypothesis class of size $r$. Let $\alpha_r, \beta_r > 0$ be such that for all $i \neq j$, and all $P_i, P_j \in \mathcal{P}_r$,*

$$d(\theta(P_i), \theta(P_j)) \geq \alpha_r;$$
$$KL(P_i, P_j) \leq \beta_r.$$

*Then*

$$\max_{j \in [r]} \mathbb{E}_{P_j}[d(\widehat{\theta}(X), \theta(P_j))] \geq \frac{\alpha_r}{2}\left(1 - \frac{\beta_r + \log 2}{\log r}\right).$$

**Definition A.25** (Relative Hamming Distance)**.** *For $\boldsymbol{x}, \boldsymbol{y} \in \{0, 1\}^m$, we define their* relative Hamming distance *as follows:*

$$d_H(\boldsymbol{x}, \boldsymbol{y}) := \frac{1}{m} |\{i \in [m] : x_i \neq y_i\}|.$$

We will need the following construction of coupled codes.

**Proposition A.26.** *Let $m_P, m_Q \geq 2$ and $m_Q$ divide $m_P$. There exists a code $C \subset \{0,1\}^{m_P}$ and a projection map $\Pi : \{0,1\}^{m_P} \to \{0,1\}^{m_Q}$ such that if $C' = \{\Pi(w) : w \in C\}$ then $C'$ is a code with relative Hamming distance $\Omega(1)$. Moreover, $|C| = |C'| \geq 2^{0.1 m_Q}$*

Throughout the proof, we will identify the community assignment function $z : [n] \to [k]$ of an SBM (Definition 3.1) with the matrix $Z \in \{0,1\}^{n \times k}$ where $Z_{ij} = 1$ if and only if $z(i) = j$.

*Proof.* Begin with a Gilbert-Varshamov code $B \subset \{0,1\}^{m_Q}$ as in Theorem A.23. We can "lift" $B$ to a code on $\{0,1\}^{m_P}$ simply by concatenation. If $w \in B$, then the corresponding $w' \in C$ is just $w' = (w, w, \ldots, w) \in \{0,1\}^{m_P}$. Let $\Pi : \{0,1\}^{m_P} \to \{0,1\}^{m_Q}$ simply select the first $m_Q$ bits of a word. It is clear that $B = \{\Pi(w) : w \in C\}$, so we are done. $\qquad\square$

Now we are ready to prove Theorem 3.2.

*Proof of Theorem 3.2.* Let $m_P = \binom{n}{2}$, $m_Q = \binom{n_Q}{2}$, and $m = m_P$. Let $C \subset \{0,1\}^{m_P}$ be the code and $\Pi : \{0,1\}^{m_P} \to \{0,1\}^{m_Q}$ the projection map of Prop A.26. For each $w \in C$, we construct a pair of SBMs $P_w, Q_w \in \mathbb{R}^{n \times n}$ as follows.

Each $P_w, Q_w$ is a stochastic block model with $k_P, k_Q$ classes respectively. All the $P_w$ share the same community structure, namely the lexicographic assignment where nodes $1, 2, \ldots, \frac{n}{k_P}$ are assigned to community 1, and so on. Similarly all the $Q_w$ share the same lexicographic community structure with nodes $1, 2, \ldots, \frac{n}{k_Q}$ assigned to community 1, and so on. Therefore, there are fixed $Z_P \in \{0,1\}^{n \times k_P}$, $Z_Q \in \{0,1\}^{n \times k_Q}$, such that for all $w \in C$, there exist $A_w \in \mathbb{R}^{k_P \times k_P}$, $B_w \in \mathbb{R}^{k_Q \times k_Q}$ with

$$P_w = Z_P A_w Z_P^T,$$
$$Q_w = Z_Q B_w Z_Q^T.$$

The $A_w, B_w$ are defined as follows. Let $i, j \in [k_P]$ and $i', j' \in [k_Q]$ be such that $i < j$ and $i' < j'$. Since $m_P = \binom{k_P}{2}$ and $m_Q = \binom{k_Q}{2}$, we can identify $(i,j)$ and $(i',j')$ with indices of $[m_P], [m_Q]$ respectively. Then for fixed $\delta_P, \delta_Q > 0$, the edge connectivity probabilities are

$$A_w(i,j) = A_w(j,i) := \begin{cases} 1/2 & \text{if } w_{ij} = 0, \\ 1/2 + \delta_P & \text{if } w_{ij} = 1; \end{cases}$$

$$B_w(i',j') = B_w(j',i') := \begin{cases} 1/2 & \text{if } \Pi(w)_{i'j'} = 0, \\ 1/2 + \delta_Q & \text{if } \Pi(w)_{i'j'} = 1. \end{cases}$$

We can set the diagonals of $A_w, B_w$ to be $1/2$ as well.

Next, let $\mathcal{P}_r$ be a family of $r = |C|$ probability measures. For fixed $w \in C$, the corresponding measure is the distribution over data $(A_P, A_Q) \in \{0,1\}^{n \times n} \times \{0,1\}^{n_Q \times n_Q}$ sampled from $(P_w, Q_w[S,S])$. Note that we restrict $S$ to be a fixed subset of $[n]$.

Next, let $\theta((P_w, Q_w)) := Q_w$, and let $d(\theta((P_w, Q_w)), \theta((P_{w'}, Q_{w'}))) := \frac{1}{n} \|Q_w - Q_{w'}\|_F$. We will show that for all $w, w' \in C$ with $w \neq w'$,

$$KL((P_w, Q_w), (P_{w'}, Q_{w'})) \leq KL(P_w, P_{w'}) + KL(Q_w, Q_{w'})$$
$$\leq O(n^2 \delta_P^2 + n_Q^2 \delta_Q^2)$$
$$=: \beta,$$
$$d((P_w, Q_w), (P_{w'}, Q_{w'})) := \frac{1}{n} \|Q_w - Q_{w'}\|_F$$
$$\geq \Omega(\delta_Q)$$
$$=: \alpha.$$

For the $\beta$ claim, by Proposition 4.2 of Gao et al. (2015), if $\delta_P, \delta_Q \in (0, 1/4)$, we have

$$KL((P_w, Q_w), (P_{w'}, Q_{w'})) \leq KL(P_w, P_{w'}) + KL(Q_w, Q_{w'})$$
$$\lesssim \sum_{i,j \in [n]} (P_w(i,j) - P_{w'}(i,j))^2 + (Q_w(i,j) - Q_{w'}(i,j))^2.$$

Next, notice that $A_w(i,j) \neq A_{w'}(i,j)$ if and only if $w_{ij} \neq w'_{ij}$. Then for distinct $w, w' \in C$, we have $d_H(w, w') = \Omega(m_P)$, so

$$\sum_{i,j \in [n]} (P_w(i,j) - P_{w'}(i,j))^2 \lesssim \delta_P^2 \frac{n^2}{k_P^2} d_H(w, w') \binom{k_P}{2} \lesssim \delta_P^2 n^2.$$

The bound for $Q_w$ is similar, so this verifies the $\beta$ claim.

Similarly, for the $\alpha$ claim, notice that

$$\frac{1}{n} \|Q_w - Q_{w'}\|_F \gtrsim \frac{1}{k_Q} \sqrt{\delta_Q^2 d_H(\Pi(w), \Pi(w'))} \geq \frac{\delta_Q}{k_Q} \sqrt{d_H(\Pi(w), \Pi(w'))}.$$

By Prop A.26, $d_H(\Pi(w), \Pi(w')) = \Omega(m_Q) = \Omega(k_Q^2)$. Therefore $\alpha \leq \Omega(\delta_Q)$.

Next, by Prop A.27, the pair $(P_w, Q_w)$ satisfies Definition 1.3 for all $w \in C$. Moreover, $\log |C| \geq 0.1 m_Q$ by Prop A.26.

Combining these results, by Theorem A.24 the overall lower bound is

$$\inf_{\widehat{Q}} \sup_w \frac{1}{n} \|\widehat{Q} - Q_w\|_F \gtrsim \alpha \left(1 - \frac{\beta + \log 2}{0.1\binom{k_Q}{2}}\right)$$

$$\geq \delta_Q \left(1 - \frac{30n^2\delta_P^2}{k_Q^2} - \frac{30n_Q^2\delta_Q^2}{k_Q^2} - o(1)\right).$$

If we choose $\delta_P = 0.01(\frac{k_Q}{n})$ and $\delta_Q = 0.01\frac{k_Q}{n_Q}$, then

$$\inf_{\widehat{Q}} \sup_w \frac{1}{n^2} \|\widehat{Q} - Q_w\|_F^2 \gtrsim \delta_Q^2$$

$$\gtrsim \frac{k_Q^2}{n_Q^2}.$$

Note that $k_Q \leq n_Q \leq n$, so $\delta_P, \delta_Q \in (0, 1/4)$ as desired. $\qquad\square$

**Proposition A.27.** *If $h_n = \min\{\frac{1}{k_P}, \frac{1}{k_Q}\}$ then for all $w \in C$, the pair $(P_w, Q_w)$ satisfies Defn 1.3 at $h_n$.*

*Proof.* Consider $h = h_n$ and some node $i \in [n]$. Suppose that $j \neq i$ is in the same $P_w$-community as $i$, and that $\ell \neq i$ is in a different community. Then notice that $d_{P_w}(i, \ell) \geq d_{P_w}(i, j)$. Therefore $j \in T_i^{P_w}(h)$. Moreover, since $h \leq \frac{1}{k_P}$ and since the nodes of $S \subset [n]$ are equidistributed among the communities $1, 2, \ldots, k_P$, it follows that all members of $T_i^{P_w}(h)$ must belong to the same $P_w$-community as $i$.

Therefore, since the communities of $Q_w$ are a coarsening of the communities of $P_w$, $j \in T_i^{Q_w}(\frac{1}{k_Q})$. Since $h \leq \frac{1}{k_Q}$, we are done. $\qquad\square$

### A.4   SBM Clustering Error

In this section, we prove a minimax lower bound in the clustering regime for stochastic block models.

**Theorem A.28.** *Let $\Pi$ denote the parameter space of pairs of SBMs $(P, Q)$ on $n$ nodes with $k_P, k_Q$ communities respectively, such that the cluster structure of $Q$ is a coarsening the cluster structure of $P$. Then*

$$\inf_{\widehat{Q}} \sup_{(P,Q)\in\Pi} \mathbb{E}[\frac{1}{n^2} \|\widehat{Q} - Q_i\|_F^2] \gtrsim \frac{\log k_Q}{n_Q}.$$

*Proof.* Let $H_m \in [0, 1]^{m \times m}$ be the Hadamard matrix of order $m$ modified to replace all entries $-1$ with 0. If $m$ is not a power of two, let $H_m$ be defined as follows. Let $\ell = \lfloor \log_2 m \rfloor$ and let $H_{m'} \in \mathbb{R}^{m/2 \times m/2}$ contain $H_{2^{\ell-1}}$ on its top left block and zeroes elsewhere. Let

$$H_m = \begin{bmatrix} \mathbf{0}\mathbf{0}^T & H_{m'} \\ H_{m'}^T & \mathbf{0}\mathbf{0}^T \end{bmatrix}.$$

Notice that at most $\frac{7}{8}$ fraction of the entries of $H_m$ are zero-padded, for any $m$. Now, let $B_P = \frac{1}{2}\mathbf{1}\mathbf{1}^T + \delta_P H_{k_P}$ and $B_Q = \frac{1}{2}\mathbf{1}\mathbf{1}^T + \delta_Q H_{k_Q}$ for some $\delta_P, \delta_Q \in (0, 1/4)$ to be chosen later.

We will define two families of matrices indexed by a finite set $T$. For $i \in T$, there are some $Z_i \in \{0, 1\}^{n \times k_P}$ and $Y_i \in \{0, 1\}^{n \times k_Q}$ to be specified later. Then

$$P_i = Z_i B_P Z_i^T,$$
$$Q_i = Y_i B_Q Y_i^T.$$

Now, we define $Y_i$ as follows. Let $Z_{n,k_Q}$ denote the set of balanced clusterings $z : [n] \to [k_Q]$ such that for all $i, j \in [k_Q]$, $\left|z^{-1}(\{i\})\right| = \left|z^{-1}(\{j\})\right|$. Let $Z \subset Z_{n,k_Q}$ select the $z$ such that for all $j \leq k_Q/2$, $z^{-1}(j) = \{\left\lfloor \frac{n(j-1)}{k_Q} \right\rfloor, \ldots, \left\lfloor \frac{nj}{k_Q} \right\rfloor\}$. Define a distance on $Z$ as follows. For $y, y' \in Z$ let $Y, Y' \in \{0,1\}^{n \times k_Q}$ be the corresponding cluster matrices and let $d(y, y') := \frac{1}{n}\|YB_QY^T - Y'B_Q(Y')^T\|_F$. By Theorem 2.2 of Gao et al. (2015), there exists a packing $T_0 \subset Z$ with respect to $d$ such that for all $y, y' \in T_0$, we have $|\{j : y'(j) \neq y(j)\}| \geq n/6$. Moreover, $\log |T_0| \geq \frac{1}{12}n \log k_Q$. Set $T = T_0$. For any $y_i \in T_0$, let $Y_i \in \{0,1\}^{n \times k_Q}$ be the corresponding cluster matrix and then $Q_i = Y_iB_QY_i^T$.

Now, to define $Z_i$, take $a \in [k_Q]$ and partition $y_i^{-1}(\{a\}) \subset [n]$ into $k_P/k_Q$ equally sized communities in a uniformly random way. Number these $1, \ldots, \frac{k_P}{k_Q}$. In this way, we split community 1 of $y_i$ into communities $1, \ldots, \frac{k_P}{k_Q}$ of $z_i$, and so on. Define $Z_i$ to be the matrix corresponding to $z_i$. Notice that $Z_i, Y_i$ are both balanced clusterings and that the clustering $Y_i$ coarsens that of $Z_i$. Therefore $(P_i, Q_i)$ are a pair of heterogeneous symmetric SBMs satisfying Definition 1.3 at $h = 1/k_Q$.

Next, we apply Fano's Inequality (Theorem A.24). Recall $\log |T| \geq \frac{1}{12}n \log k_Q$. Now, for $i, j \in T$ distinct, Prop 4.2 of Gao et al. (2015) gives

$$D_{KL}((P_i, Q_i), (P_j, Q_j)) \leq D_{KL}(P_i, P_j) + D_{KL}(Q_i, Q_j) \leq O(n^2\delta_P^2 + n_Q^2\delta_Q^2) =: \gamma_1.$$

Finally, we can bound:

$$\frac{1}{n^2}\|Q_i - Q_{i'}\|_F^2 \geq \frac{1}{n^2} \sum_{n/2 < j \leq n} \frac{n}{k_Q}\|(e_{y_i(j)} - e_{y'_i(j)})B_Q\|^2$$

$$\geq c_0\delta_Q^2 =: \gamma_2^2,$$

where $c_0 > 1$ is some constant. This follows because there are a constant fraction of $j > n/2$ such that $y_i(j) \neq y'_i(j)$, and any two rows of the Hadamard matrix differ on half their entries.

Now, set $\delta_Q^2 = \frac{n_Q \log k_Q}{10n_Q^2}$ and $\delta_P^2 = \frac{\log k_Q}{10n^2}$. Since $n \geq n_Q$, we conclude that

$$\inf_{\widehat{Q}} \sup_{i \in T} \mathbb{E}\left[\frac{1}{n^2}\|\widehat{Q} - Q_i\|_F^2\right] \gtrsim \gamma_2^2\left(1 - \frac{\gamma_1 + \log 2}{(1/12)n \log k_Q}\right)$$

$$\gtrsim \frac{\log k_Q}{n_Q}. \qquad \square$$

## A.5 Proof of Proposition 3.4

We first argue that Algorithm 2 perfectly recovers $Z_P, Z_Q$ with high probability.

**Theorem A.29** (Implicit in Chen et al. (2014)). *Let $M = ZBZ^T$ be an $(n, n_{min}, s)$-HSBM. Then there exists absolute constant $C > 0$ such that the Algorithm of Chen et al. (2014) can recover $Z$, up to permutation, with zero error with probability $\geq 1 - O(n^{-8})$ if*

$$s \geq C\left(\frac{\sqrt{n}}{n_{\min}} \vee \frac{\log^2(n)}{\sqrt{n_{\min}}}\right).$$

*Proof.* The algorithm of Chen et al. (2014) returns a matrix $Y \in \{0,1\}^{n \times n}$ such that $Y_{ij} = 1$ if and only if $i, j$ are in the same community, with probability $\geq 1 - O(n^{-8})$. Therefore, to construct a clustering from $Y$, simply assign the cluster of node 1 to all $j \in [n]$ such that $Y_{1j} = 1$, and so on. This returns the true $Z \in \{0,1\}^{n \times k}$ up to permutation with probability $\geq 1 - O(n^{-8})$. Note that $k$ is correctly chosen because $Y$ is equal to a block-diagonal matrix of ones up to permutation, with $k$ blocks. $\qquad \square$

Theorem A.29 implies the following.

**Proposition A.30.** *Let* $\widehat{Z}_P, \widehat{Z}_Q$ *be as in Algorithm 2. Let* $s_P, s_Q$ *be the signal to noise ratios of* $P, Q$ *respectively. If* $s_P, s_Q$ *satisfy the conditions of Theorem A.29 with respect to* $(n, n_{\min}^{(P)}$ *and* $(n_Q, n_{\min}^{(Q)})$ *respectively, then then with probability* $\geq 1 - O(n_Q^{-8})$, *there are permutation matrices* $U_P \in \{0,1\}^{k_P \times k_P}, U_Q \in \{0,1\}^{k_Q \times k_Q}$ *such that* $\widehat{Z}_P = Z_P U_P$ *and* $\widehat{Z}_Q = Z_Q U_Q$.

Next, we want to recover the clustering of $Q$ on all $n$ nodes, not just the $n_Q$ nodes that we observe in $A_Q$. This is given by the following.

**Proposition A.31.** *1. If* $h_n = 1/k_P$ *and* $k_Q \leq k_P$ *then there exists a unique* $\Pi \in \{0,1\}^{k_P \times k_Q}$ *such that* $Z_P \Pi$ *contains the Q-clustering of all nodes in* $[n]$. *Let* $\widetilde{Z}_Q := Z_P \Pi$.

*2. Let* $\widehat{\Pi}$ *be as in Algorithm 2 and* $U_P, U_Q$ *be as in Proposition A.30. Then with probability* $1 - O(\frac{1}{n_Q})$, $Z_P U_P \widehat{\Pi} = \widetilde{Z}_Q U_Q$.

*Proof.* Part (1) follows immediately from the SBM structure of $P, Q$ and definition of Definition 1.3. For Part (2), first notice that by Proposition A.30, with probability at least $1 - O(\frac{1}{n_Q})$, Algorithm 2 returns the true clusterings $\widehat{Z}_P = Z_P \in \{0,1\}^{n \times k_P}$ and $\widehat{Z}_Q = Z_Q \in \{0,1\}^{n_Q \times k_Q}$, up to permutation.

Now, Algorithm 2 simply takes unions of the clusters of $Z_P$ to learn $\widehat{\Pi}$. Therefore, let $V : \mathbb{R}^n \to \mathbb{R}^{n_Q}$ project onto coordinates in $S$. Then $V \widehat{Z}_P \widehat{\Pi} = \widehat{Z}_Q$. Moreover, by Proposition A.30, $\widehat{Z}_P = Z_P U_P$ and $\widehat{Z}_Q = Z_Q U_Q$. Hence $V Z_P U_P \widehat{\Pi} = V \widetilde{Z}_Q U_Q$. To remove dependence on $V$, we need to argue that each $Q$-cluster has a reprensentatve in $S$.

Let $E$ be the event that at least one $Q$-cluster has no representative in $S$. For a fixed $j \in [k_Q]$, cluster $j$ has no representative in $S$ with probability $\leq \left(1 - \frac{n_{\min}^{(Q)}}{n_Q}\right)^{n_Q}$. A union bound implies that

$$\mathbb{P}[E] \leq k_Q \left(1 - \frac{n_{\min}^{(Q)}}{n_Q}\right)^{n_Q} \leq k_Q \exp(-n_{\min}^{(Q)}) \leq O(n_Q^{-1}).$$

The last inequality holds because the condition of Theorem A.29 implies that $n_{\min}^{(Q)} \geq \Omega(\sqrt{n_Q})$ and $k_Q \leq \frac{n_Q}{n_{\min}^{(Q)}}$.

Finally, we proceed by conditioning on $\neg E$. Since $\widehat{Z}_P = Z_P U_P$, we know that for all $i \in S$, the unique $j_P \in [k_P]$ such that row $i$, column $j_P$ of $Z_P$ is nonzero contains its true $P$-community up to $U_P$. Similarly since $\widehat{Z}_Q = Z_Q U_Q$, the the unique $j_Q \in [k_Q]$ such that row $i$, column $j_Q$ of $Z_P$ is nonzero contains its true $Q$-community up to $U_Q$. Therefore the nodes in community $j_P$ in $P$ are in community $j_Q$ in $Q$. So, up to permutations $U_P$ and $U_Q$, we have $\Pi_{j_P, j_Q} = 1$. Since we condition on $\neg E$, each cluster of $Q$ has at least one representative in $S$, so each columns of $\Pi$ is nonzero. We conclude that $Z_P U_P \widehat{\Pi} = \widetilde{Z}_Q U_Q$ with probability at least $1 - O(n_Q^{-1})$. $\square$

We are ready to give the overall error of Proposition 2.

**Proposition A.32.** *Suppose that* $\widehat{Z}_P = Z_P, \widehat{\Pi} = \Pi$ *in Algorithm 2. Then with probability* $\geq 1 - O(\frac{1}{n_Q})$, *Algorithm 2 returns a* $\widehat{Q} \in [0,1]^{n \times n}$ *such that*

$$\frac{1}{n^2} \|\widehat{Q} - Q\|_F^2 \lesssim \frac{k_Q^2 \log(n_{\min}^{(Q)})}{n_Q^2}.$$

*Proof.* By Proposition A.31, with probability $\geq 1 - O(\frac{1}{n_Q})$, we have $\widehat{Z}_P = Z_P U_P$, $\widehat{Z}_Q = Z_Q U_Q$, and $\widetilde{Z}_Q U_Q = Z_P U_P \widehat{\Pi}$. We proceed by conditioning on these events.

Next, let $W_Q \in \mathbb{R}^{k_Q \times k_Q}$ be the population version of $\widehat{W}_Q$ with $W_{Q;ii} = (\mathbf{1}^T Z_Q e_i)^{-1}$. Then since $\widehat{Z}_Q = Z_Q U_Q$ we have $\widehat{W}_Q = U_Q^T W_Q U_Q$. Hence

$$
\begin{aligned}
\widehat{Q} &= (Z_P U_P \widehat{\Pi})(U_Q^T W_Q U_Q^T)(Z_Q U_Q)^T A_Q (Z_Q U_Q)(U_Q^T W_Q U_Q)(Z_P U_P \widehat{\Pi})^T \\
&= \widetilde{Z}_Q (W_Q Z_Q^T A_Q Z_Q W_Q) \widetilde{Z}_Q^T.
\end{aligned}
$$

Next, let $z_Q : [n] \to [k_Q]$ be the ground truth clustering map given by $\widetilde{Z}_Q \in \{0,1\}^{n \times k_Q}$. Let $B_Q$ be defined analogously to $\widehat{B}_Q$ in Algorithm 2, but using $W_Q, Z_Q, \mathbb{E}[A_Q]$ in place of $\widehat{W}_Q, \widehat{Z}_Q, A_Q$. Let $m_i := W_{Q;ii}^{-1}$ be the the number of nodes in $S$ belong to community $i$, and let $n_i$ be the the number of nodes in $[n]$ belonging to community $i$ of $Q$. Then the error of Algorithm 2 is then

$$
\begin{aligned}
\frac{1}{n^2}\|\widetilde{Z}_Q(\widehat{B}_Q - B_Q)\widetilde{Z}_Q^T\|_F^2 &= \frac{1}{n^2}\left( \sum_{i,j\in[k_Q]} n_i n_j \left( \sum_{\substack{r\in z_Q^{-1}(\{i\})\cap S \\ s\in z_Q^{-1}(\{j\})\cap S}} \frac{B_{Q;ij} - A_{Q;rs}}{m_i m_j} \right)^2 \right) \\
&= \frac{1}{n^2}\sum_{i,j\in[k_Q]} \frac{n_i n_j}{m_i^2 m_j^2}\left( \sum_{\substack{r\in z_Q^{-1}(\{i\})\cap S \\ s\in z_Q^{-1}(\{j\})\cap S}} B_{Q;ij} - A_{Q;rs} \right)^2.
\end{aligned}
$$

Next, fix $i,j \in [k_Q]$ and let

$$
X_{ij} = \sum_{\substack{r\in z_Q^{-1}(\{i\})\cap S \\ s\in z_Q^{-1}(\{j\})\cap S}} B_{Q;ij} - A_{Q;rs}.
$$

If we condition on the clusterings of $P, Q$ being correct then $\mathbb{E}[B_{Q;ij} - A_{Q;rs}] = 0$. Therefore by Hoeffding's inequality,

$$
\mathbb{P}(X_{ij} \geq t^2) \leq 2\exp\left( -\frac{2t^2}{m_i m_j} \right).
$$

Setting $t^2 = 10\log(m_i m_j) m_i m_j$ implies that with probability at least $1 - k_Q^2 \min_i(m_i)^{-20}$, that the overall error is

$$
\frac{1}{n^2}\|\widehat{Q} - Q\|_F^2 \leq \frac{1}{n^2}\sum_{i,j\in[k_Q]} \frac{10\log(m_i m_j) n_i n_j}{m_i m_j}.
$$

Finally, note that there exists a constant $c_0 > 0$ such that for all $i \in [k_Q]$, $m_i \geq c_0\sqrt{n_Q}$ and $n_i \geq c_0\sqrt{n}$, by assumption. Note that each $m_i$ is a random quantity depending on the choice of $S \subset [n]$ such that $\mathbb{E}[m_i] = \frac{n_Q}{n}n_i$. Hoeffding's inequality and a union bound over all $i \in [k_Q]$ imply that that with probability at least $\geq 1 - O(n_Q^{-8})$ that $m_i \geq \mathbb{E}[m_i] - 10\sqrt{\log n_Q} \geq \Omega(\mathbb{E}[m_i])$. We conclude that

$$
\begin{aligned}
\frac{1}{n^2}\|\widehat{Q} - Q\|_F^2 &\leq O\left( \frac{1}{n_Q^2}\sum_{i,j\in[k_Q]} 10\log(m_i m_j) \right) \\
&\leq O\left( \frac{k_Q^2 \log(n_{\min}^{(Q)})}{n_Q} \right). \qquad \square
\end{aligned}
$$

# B  Additional Experiments

## B.1  Ablation Experiments

In this section, we discuss additional experiments that quantify the dependence of our algorithms on all relevant parameters. Our experiments also include a new baseline adapted from the estimator of Levin et al. (2022).

**Description of New Baseline.** Levin et al. (2022) assumes that full edge data from both P and Q are observed, and $P = Q$. Since this is not true for us, we instead compute the following modified MLE based on their estimator from Section 3.3 of Levin et al. (2022).

$$\widetilde{Q}_{ij} = \begin{cases} \frac{w_P}{w_P + w_Q} A_{P;ij} + \frac{w_Q}{w_P + w_Q} A_{Q;ij} & \text{if } i, j \in S, \\ A_{P;ij} & \text{otherwise.} \end{cases}$$

Here $w_P, w_Q$ are computed as in their paper, based on estimated sub-gamma parameters of the noise for $A_P, A_Q$. Akin to their adjacency spectral embedding, which assumes known rank of $Q$, we use Universal Singular Value Thresholding to obtain $\widehat{Q}$ from $\widetilde{Q}$ Chatterjee (2015).

**Oracle with $p = 0.0$.** In addition to testing the new baseline from Levin et al. (2022), we also test the Oracle baseline with $p = 0.0$. As noted in Section 4, this corresponds to the non-transfer setting where all edges from the target graph $Q$ are observed. Note that in this case, the value of $n_Q$ does not matter because edges incident to nodes outside of $S$ never get flipped. The Oracle error for $\beta$-smooth graphons on $d$-dimensional latent variables will therefore be $O(n^{-\frac{2\beta}{2\beta+d}})$ Xu (2018), which is less than the error bound of Theorem 2.3. Indeed, we will find that the Oracle our transfer algorithms in the regimes where its theoretical upper bound is better than our theoretical upper bounds.

Next, we describe the experimental results.

Figure 3 tests Algorithm 1 for general latent variable models. The error (Theorem 2.3) depends on the smoothness $\beta$ of the target graph, the number of observed target nodes $n_Q$, and the dimension of the latent variables $d$.

Figure 4 tests Algorithm 2 for Stochastic Block Models. The error (Proposition 3.4) depends on the number of communities $k_Q$ in the target graph, and the number of observed target nodes $n_Q$. Note that Proposition 3.4 also depends logarithmically on the minimum community size of $Q$, but this is less significant.

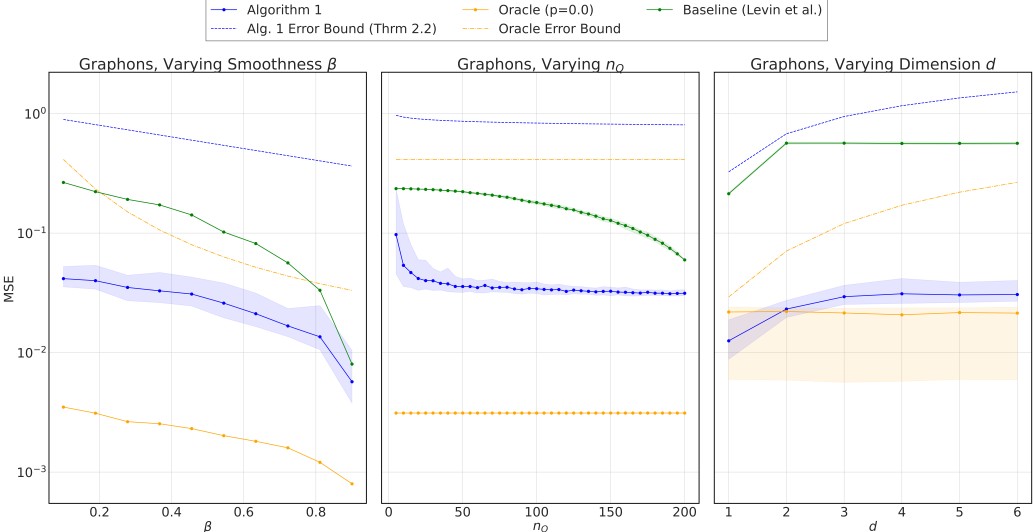

Figure 3: Testing parameters of Algorithm 1 (Transfer for Latent Variable Models). For most parameter settings, our method is better than the baseline and worse than the Oracle.
**Left**: Testing Hölder-smoothness of $f_Q$ with $n = 200, n_Q = 25, d = 1$. All methods improve as $\beta \to 1$. Here $f_P(x, y) = \frac{x^\alpha + y^\alpha}{2}, f_Q(x, y) = \frac{x^\beta + y^\beta}{2}$ with $\alpha = 0.01$ and $\beta$ varying.
**Middle**: Testing number of observed target nodes $n_Q$ with $n = 200, d = 1$. Here $f_P(x, y) = \frac{x^\alpha + y^\alpha}{2}, f_Q(x, y) = \frac{x^\beta + y^\beta}{2}$ with $\alpha = 0.01, \beta = 0.1$. Note that the oracle does not depend on $n_Q$ because it observes the full adjacency matrix $A_Q \in \{0, 1\}^{n \times n}$.
**Right**: Testing dimension $d$ of latent positions $\boldsymbol{x}_1, \ldots, \boldsymbol{x}_n \in [0, 1]^d$ (i.i.d. Lebesgue) with $n = 200, n_Q = 25$. Here $f_P(\boldsymbol{x}, \boldsymbol{y}) = \exp(-6\|\boldsymbol{x} - \boldsymbol{y}\|_2)$ and $f_Q(\boldsymbol{x}, \boldsymbol{y}) = \exp(-|x_1 - y_1|)$.
Points are the median MSE across 50 trials, with with $[5, 95]$ percentile outcomes shaded.

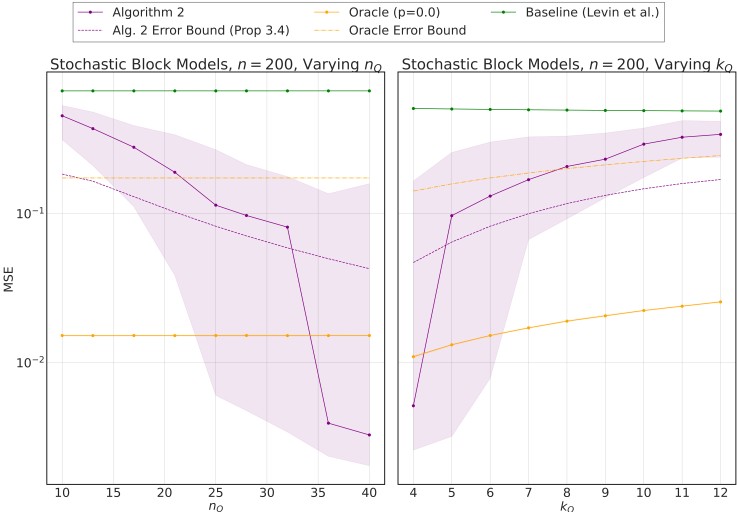

Figure 4: Testing parameters of Algorithm 2 (Transfer for SBMs). For most parameter settings, our method is better than the baseline and worse than the Oracle.
**Left**: $n = 200, k_P = 12, k_Q = 6$. Note that the oracle does not depend on $n_Q$ because it observes the full adjacency matrix $A_Q \in \{0,1\}^{n \times n}$.
**Right**: $n = 200, n_Q = 25, k_P = 2k_Q$.
For both experiments, the intra-community edge probabilities are $0.2, 0.9$ for $P, Q$ respectively, while the inter-community edge probabilities are $0.1, 0.8$ respectively. Points are the median MSE across $50$ trials, with with $[5, 95]$ percentile outcomes shaded.

Note that while we can plot theoretical guarantees for the mean squared error $\frac{1}{n^2}||\widehat{Q} - Q||_F^2$ of both our algorithms' $\widehat{Q}$ and the oracle's $\widehat{Q}$, Levin et al. (2022) only give theoretical guarantees on the spectral norm $||\widehat{Q} - Q||_2$ for their estimator $\widehat{Q}$. Analyzing the stronger metric of mean-squared error would require different techniques than their paper.

## B.2 Link Prediction Experiments

In this section, we present additional link prediction experiments on the EMAIL-EU and BIGG MODELS datasets. Unlike Section 4, we tune the sparsity estimate $\widehat{\rho} \in (0, 1)$ used in the Universal Singular Value Thresholding step of the Oracle baseline. In particular, we set $\widehat{\rho} \in (0, 1)$ to be the mean of the entries of the ground truth target matrix $Q \in [0, 1]^{n \times n}$. Note that this value is inaccessible to other algorithms since it requires knowing all the edges of $Q$.

Figures 7 and 8 show the performance of our Algorithms on the EMAIL-EU dataset, and Figures 5 and 6 for the BIGG MODELS dataset. As in the mean-squared error setting (Figure 2), we find that Algorithm 1 outperforms Algorithm 2, and that the Oracle baseline outperforms both for small $p$. Moreover, we find that the choice of source & target affects the performance of both of our algorithms. Hence Figure 7 shows better performance than Figure 8 for the same source but different targets, and Figure 5 shows better performance than Figure 6 for the same target but different sources.

## C  Experimental Details

In this section, we give further details on the experiments of Section 4.

**Compute Environment.** We run all experiments on a personal Linux machine with 378GB of CPU/RAM. The total compute time across all results in the paper was less than 2 hours.

**Functions for Figure 1.** For the top row, the source is an $(n, 4)$-SBM with $0.8$ on the diagonal and $0.2$ on the off-diagonal of $B \in \mathbb{R}^{4 \times 4}$. The target is an $(n, 2)$-SBM with $0.9$ on the diagonal and $0.1$ on the off-diagonal of $B \in \mathbb{R}^{2 \times 2}$.

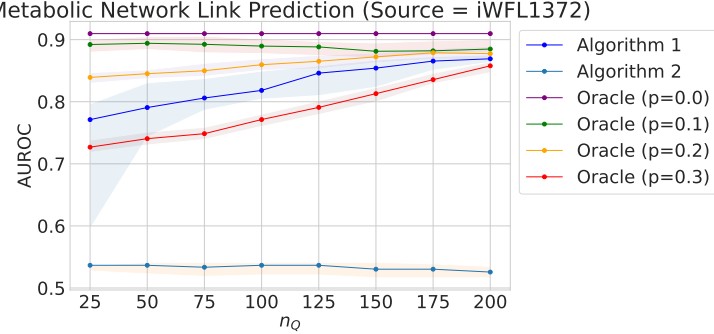

Figure 5: Link prediction results with the metabolic network of BiGG model iWFL1372 (*Escherichia coli W*) as the source and iJN1463 (*Pseudomonas putida*) the target. Shaded regions denote $[5, 95]$ percentile outcomes from $50$ independent trials.

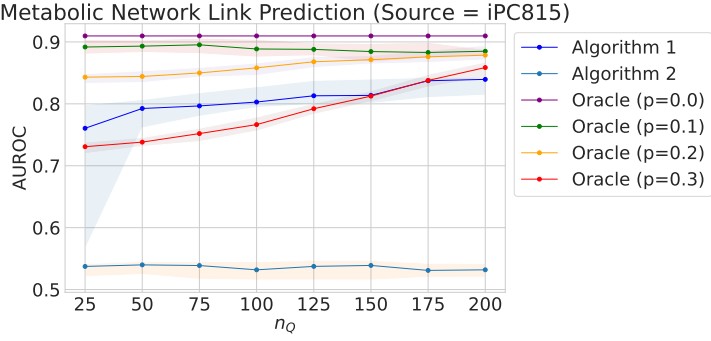

Figure 6: Link prediction results with the metabolic network of BiGG model iPC815 (*Yersinia pestis*) as the source and iJN1463 (*Pseudomonas putida*) the target. Shaded regions denote $[5, 95]$ percentile outcomes from $50$ independent trials.

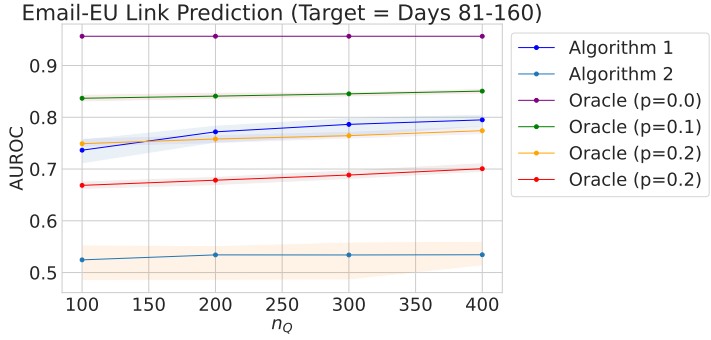

Figure 7: Link prediction results with Days 1-80 of EMAIL-EU as the source, and Days 81-160 as target. Shaded regions denote $[5, 95]$ percentile outcomes from $50$ independent trials.

For the second and third rows, the source function is $Q(x, y) = \frac{1 + \sin(\pi(1 + 3(x + y - 1)))}{2}$ (modified from Zhang et al. (2017)). The sources are $P(x, y) = 1 - Q(x, y)$ and $P(x, y) = Q(\phi(x), y)$, where $\phi(x) = 0.5 + |x - 0.5|$ if $x < 0.5$, and $0.5 - |x - 0.5|$ otherwise.

**Metabolic Networks.** We access metabolic models from King et al. (2016) at `http://bigg.ucsd.edu`. To construct a reasonable set of shared metabolites across the networks, we take the intersection of the node sets for the following BiGG models: iCHOv1, IJN1463, iMM1415, iPC815, iRC1080,

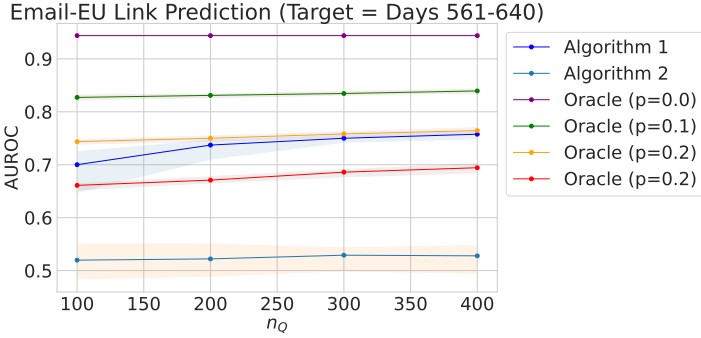

Figure 8: Link prediction results with Days 1-80 of EMAIL-EU as the source, and Days 561-640 as target. Shaded regions denote $[5, 95]$ percentile outcomes from 50 independent trials.

iSDY1059, iSFxv1172, iYL1228, iYS1720, and Recon3D. We obtain a set of $n = 251$ metabolites that are present in all of the listed models.

The resulting networks are undirected, unweighted graphs on 251 nodes. We construct the matrix $A_P \in \{0, 1\}^{n \times n}$ for species $P$ by setting $A_{P;uv} = 1$ if and only if $u$ and $v$ co-occur in a metabolic reaction in the BiGG model for $P$.

**EMAIL-EU.** We use the "email-EU-core-temporal" dataset at `https://snap.stanford.edu/data/email-Eu-core-temporal.html`, as introduced in Paranjape et al. (2017). Note that we do not perform any node preprocessing, so we use all $n = 1005$ nodes present in the data, as opposed to Leskovec and Krevl (2014); Paranjape et al. (2017) who use only 986 nodes.

Data consist of triples $(u, v, t)$ where $u, v$ are anonymized individuals and $t > 0$ is a timestamp. We split the data into 10 bins based on equally spaced timestamp percentiles. For simplicity we refer to these time periods as consisting of 80 days each in Section 4, but technically there are 803 days total. The network at time period $\ell$ consists of an unweighted undirected graph with adjacency matrix entry $A_{uv} = 1$ if and only if $(u, v, t)$ or $(v, t, u)$ occurred in the data for an appropriate timestamp $t$.

**Hyperparameters.** We do not tune any hyperparameters. For Algorithm 1 we use the quantile cutoff of $h_n = \sqrt{\frac{\log n_Q}{n_Q}}$ in all experiments.

