# OpenReview forum: "Transfer Learning for Latent Variable Network Models"
_NeurIPS.cc/2024/Conference — NeurIPS 2024 poster_

### Official Review · Reviewer_TFsn · 2024-07-01

**Soundness:** 4
**Presentation:** 4
**Contribution:** 4
**Rating:** 8
**Confidence:** 5

**Summary:**

This paper studies the problem of transfer learning for estimating the edge probabilities of random graphs under latent position models. In particular, given a fully observed graph on $n$ nodes generated from an independent-edge random graph model, with edge probabilty matrix $P$, and an $n_Q \times n_Q$ submatrix of a graph generated from an independent-edge random graph with edge probability matrix $Q$, the authors consider the task of estimating $Q$, by leveraging information from both graphs. In order for transfer learning to be possible, the underlying graph models must share some structure. The authors assume that both $P$ and $Q$ are given by latent positions models, i.e.

$$
P_{ij} = f_P(x_i,x_j),\qquad Q_{ij} = f_Q(x_i, x_j)
$$

where  $x_1,\ldots,x_n \in \mathcal{X} \subset \R^d$ are a common set of latent positions, drawn i.i.d. from a uniform distribution on $\mathcal{X}$, and that $f_P, f_Q$ are $\alpha$​-Hölder-smooth symmetric functions. The authors quantify the similarity between two nodes in a graph model using an interpretable distance based on their expected number of common neighbours (Definition 1.2), and measure the distance between two graph models on the same node set by comparing the ranks of these distances for each node in the graphs.

In Section 2, a simple algorithm (Algorithm 1) is proposed which estimates $Q_{ij}$ by averaging entries of $A_Q$ which are similar in terms of graph distance to $(i,j)$ in $A_P$, and under some reasonable conditions, a high-probability tail bound from the Frobenius norm difference between $\hat Q$ and $Q$ is dervived (Theorem 2.2).

In Section 3, information theoretic lower bounds for the problem are derived under the special case that $A_P$ and $A_Q$ are stochastic block model (SBM) graphs (Theorem 3.2), which are proved using Fano's lemma. An alternative algorithm (Algorithm 2) for estimating $Q$ under the SBM modelling assumption is proposed, based on the same ideas as Algorithm 1, and a high-probability tail bound (Proposition 3.4), analagous to Theorem 2.2, is derived which matches the information-theoretic lower bound up to a log factor.

In Section 4, Algorithms 1 and 2 are compared with an oracle method on a selection of simulated and real-world datasets.

Complete proofs of all theoretical results are provided in the appendix.

**Strengths:**

- This paper was a joy to read. The problem of transfer-learning on graphs is well-motivated with real-world applications and the relevant literature is reviewed extensively. The reader is guided through each aspect of the problem; every definition feels motivated and mathematical statements are followed by intuition and illustrative examples. The paper is well organised, the writing is exemplary, and proofs are complete, accurate and clear.

- This paper addresses a problem which is of great relevance to the machine-learning community, and yet provides what appears to be the first provably consistent algorithm for this problem. The proposed solution, is simple and elegant, and the derived high-probability tail bound and its proof provide a lens into the fundamentally important aspects of this problem. The lower bounds derived in Section 3 show the, despite the simplicity of the method, it is optimal (in the minimax sense) under the SBM, albeit under fairly strong signal-to-noise assumptions (see weaknesses).

- The simulated and real data experiments in Section 4 are informative and demonstrate the effectiveness of the approach.

**Weaknesses:**

- One weakness of this paper, is that their setup only considers graphs whose expected node degrees grow linearly in the number of nodes in the graph. In practice, many networks are sparse and it is common in the literature to consider asymptotic regimes in which expected node degrees grow sub-linearly. However, there is sufficient novelty in this paper that I don't see that this can be held against it. The authors claim in their conclusion that they believe their algorithm will work for when node degree are of the order $\Theta(n^{1/2})$, but not down to $\Theta(\log(n))$ (see Questions), but it is not clear *why* they believe this.
- It is not clear to me what is being shown in Figure 1. I was left quite confused. Some additional explanation here would be helpful.

**Questions:**

My only questions to the authors relate to addressing to points raised in the Weaknesses section. I should mention that the first point is a curiosity and I believe it is only the second point that needs addressing before this work is ready for publication.

**Typos:**
- I believe there is a typo in the conclusion, and that the big-Os relating to edge densities should be big-$\Omega$s. Is this correct?
- The first letter on line 111 should be capitalised.

**Limitations:**

The authors are upfront the main limitations of their work, although I think it would be helpful to mention the first weakness I mention explicitly.

---

> ### Author Rebuttal · Authors · 2024-08-06
>
> We sincerely thank the reviewer for their very encouraging comments concerning the novelty of our theoretical guarantees, the great relevance of the problem we consider, and the quality of our writing. We address their comments in detail below.
>
> ### **Why we believe our Algorithm 1 will work for expected node degree $\\tilde\\Theta(n^{1/2})$.**
>
> 1) _Graph distance concentration_. The proof of Theorem 2.2 requires that the empirical graph distance we use in Algorithm 1 concentrates around the population graph distance (Proposition A.13, line 602). We show that the concentration is strong enough for our purposes when the expected node degree grows linearly in the number of nodes. However, the common neighbors metric has also been used for network-assisted covariate estimation (Mao et. al. 2021) with expected node degree $\\tilde\\Omega(n^{1/2})$. Therefore, we expect that the analysis of Algorithm 1 would still go through for expected node degree $\\tilde\\Theta(n^{1/2})$.
>
> 2) _Empirical results on sparse real-world graphs_. We test our Algorithm 1 on sparse real-world graphs in Section 4, lines 271-287. The metabolic networks have median node degree $\approx 0.06 n$ and the email networks have median node degree $\approx 0.007 n$ (see Table 2). The performance of Algorithm 1 on these graphs suggests that the assumption of $\Theta(n)$ node degree is perhaps not necessary.
>
> ### **Why we believe our Algorithm 1 will not work for expected node degree $\Theta(\log(n))$.**
>
> For expected degree $o(n^{1/2})$, the common neighbors graph distance (Definition 1.2) fails to concentrate. However, one can modify the common neighbors distance so that it concentrates for expected node degree $\\tilde\\Omega(n^{1/3})$ (Mao et al. 2021). While this is still far from the $\Theta(\log(n))$ regime, it suggests that variations on the graph distance might ensure our Algorithm 1 works for sparser graphs.
>
> Note that for a Stochastic Block Model with two communities, there are information theoretic limitations to exact recovery when the expected node degree of the SBM is $o(\\log n)$ (see [1], Theorem 13). Latent variable models are much more general than SBMs, so it is unlikely that we can give a consistent algorithm for expected node degree $o(\\log n)$ in our setting.
>
> ### **Clarification of Figure 1.**
>
> The intention of Figure 1 is to give a visual idea of the inputs and outputs for the Algorithms we consider in our paper. It is styled after a similar figure in Zhang et al. (Biometrika, 2017).
>
> The Figure intends to show, at a glance, that Algorithms 1 and 2 both work well on Stochastic Block Models, that only Algorithm 1 works well on graphons, and that the Oracle performs well in all cases.
>
> To clarify, each row of the figure corresponds to a different source/target pair $(P, Q)$. For a fixed row, the upper triangular part on columns 2, 3, 4 corresponds a $\hat Q$ for a different algorithm. The upper triangular part of column 1 shows the true $P$. The lower triangular part of columns 1, 2, 3, and 4 is identical for a fixed row, and shows the true $Q$.
>
> For example, the first row of Figure 1 shows a pair $(P, Q)$ of Stochastic Block Models. From the left-most cell, we can see that $Q$ has $2$ communities while $P$ has $4$. The upper triangles of columns 2, 3, 4 show  $\hat Q$ for Algorithms 1, 2 and the Oracle respectively. The lower triangles in columns 1, 2, 3, 4 are all the same, and show the true edge probability matrix $Q$. Each cell displays the true $Q$ in the lower triangle for comparison.
>
> We will rewrite the Figure caption in the revision to avoid confusion.
>
> ### **Typographical errors.**
>
> The reviewer is correct that the occurrences of $O(\cdot)$ should be replaced with $\Omega(\cdot)$ in the Conclusion, lines 293 and 294. We will fix this and all other typographical errors in the revision.
>
> [1] Abbe, Emmanuel. "Community detection and stochastic block models: recent developments." Journal of Machine Learning Research 18.177 (2018): 1-86.

---

> > ### Comment · Reviewer_TFsn · 2024-08-09
> >
> > I thank the authors for their response and I continue for strongly support this paper.
> >
> > The discussion on the concentration of different graph distances is very interesting and I would welcome seeing it in the paper. This work could motivate others to develop graph distances which work at lower sparsity levels.

---

> > > ### Author Response · Authors · 2024-08-10
> > >
> > > Dear reviewer TFsn,
> > > Thank you very much for your encouraging words and your strong support. We will include additional discussion on concentration of graph distances in the revision.

---

### Official Review · Reviewer_BaBd · 2024-07-09

**Soundness:** 3
**Presentation:** 3
**Contribution:** 3
**Rating:** 7
**Confidence:** 2

**Summary:**

In this paper, the authors address two topics  in random network/graph models:
1. The transfer learning in latent variable network models. It proposed estimate of the distribution of target network from source network using a defined graph distance.
2. It also proves a minimax lower bound for Stochastic Block Models and show that a simple algorithm achieves this rate.

**Strengths:**

This is a theoretical paper that addresses the transfer learning from source to target network in latent variable network models.

**Weaknesses:**

1. Theorem 1.1 is an informal form of Theorem 2.2, it seems there is no need to repeat.
2. The authors repeatedly use the vague word "suitable", "suitably" in the paper. Many phrases (containing this word) need to be clarified.

**Questions:**

In Page 3, line 105: the authors mentioned: relative, not absolute graph distances,  it seems to me this is not defined.

**Limitations:**

In the authors' reply to the question set, in line 950, the authors mentioned that they discussed  "the need for a different graph distance ..."
. It will benefit the readers if the authors explain further in this paper on the selection of of the distance, an explain why they prefer this graph distance that they are currently using.

---

> ### Author Rebuttal · Authors · 2024-08-05
>
> We sincerely thank the reviewer for their feedback and for assessing our paper to have high impact. We address their comments in detail below.
>
> ### **Clarification of relative versus absolute graph distances.**
>
> On line 105, we note that our rankings assumption (Definition 1.3) concerns relative, not absolute graph distances. The reviewer is right to point out that the terms "relative" and "absolute" can be clarified.
>
> Our point is that Definition 1.3 involves quantiles of graph distances. This is a "relative" condition, because it depends on a rank-ordering within both graphs $P, Q$ before comparison. On the other hand, an "absolute" condition would require that for nodes $i, j \in [n]$, if e.g. $d_P(i,j) < 100$ then $d_Q(i,j) < C \cdot 100$. Our condition is more flexible and will hold for a larger set of graph pairs $(P, Q)$, such as pairs where one graph is much more dense than the other.
>
> In the revision, we will rewrite line 105 to clarify this point.
>
> ### **What we mean by "the need for a different graph distance" in line 950.**
>
> In line 950 we mention our discussion of extending to sparse graphs in the Conclusion (lines 292-295). The concentration of the empirical graph distance (Algorithm 1, line 3) requires expected degree $\tilde\Omega(n^{1/2})$ (Mao et al. 2021). In the same paper, they show that a modification of this graph distance concentrates when expected degree is $\tilde\Omega(n^{1/3})$.
>
> While this is still far from the $\Theta(\log n)$ degree regime, it suggests that variations on the graph distance might ensure our Algorithm 1 works for sparser graphs.
>
> We will clarify this point in the revision.
>
> ### **Why we select our graph distance.**
>
> We use the common neighbors graph distance (Definition 1.2) to capture local graph structure, as discussed in lines 90-100. Other graph distances have been used in the literature as well (Zhang et al. 2017, Mukherjee and Chakrabarti 2019, Mao et al. 2021).
>
> The reason we use this specific graph distance is technical. We upper-bound part of the smoothing error of Algorithm 1 in terms of the common neighbors graph distance (section A.2.3). Roughly speaking, there is a relationship between the $\ell_2$ (Euclidean) distance between rows of the common neighbors matrix $Q^2$ (coming from our graph distance), and the $\ell_2$ distance between rows of $\hat Q - Q$ (coming from the mean-squared error $\frac{1}{n^2} \|| \hat Q - Q \||_F^2$).  See Lemma A.17, lines 641-646, for a precise statement.
>
> To minimize mean-squared error, we use a graph distance based on $\ell_2$ distance. Previous works indicate that bounding other kinds of error, such as $\|| \hat Q - Q \||_{2 \to \infty}$ (Zhang et al. 2017), require graph distances tailored for those errors.
>
> We will include this point in the revision.
>
> ### **Writing clarity and typographical errors.**
>
> We will remove all uses of vague language and typographical errors in the revision.

---

> > ### Comment · Reviewer_BaBd · 2024-08-09
> >
> > Thank you for your response! I read your rebuttal and other reviewers comments and still want to keep my rating as  7: Accept.

---

> > > ### Author Response · Authors · 2024-08-10
> > >
> > > Dear reviewer BaBd,
> > > Thank you very much for your support.

---

### Official Review · Reviewer_VhmF · 2024-07-11

**Soundness:** 2
**Presentation:** 3
**Contribution:** 2
**Rating:** 6
**Confidence:** 1

**Summary:**

The work explores transfer learning in latent variable network models. In particular the work focuses on the setting of observing samples from an n x n probability matrix from a source P and a submatrix of the adjacency of a target Q. The goal is then to estimate Q, using information from P.

The authors propose an algorithm with vanishing error under certain conditions and a more simple algorithm for the special case of stochastic block models.

**Strengths:**

The paper is well-written. I particularly enjoyed the introductory sections that help motivate the work. I believe that the problem the authors tackle has value, as it stands mostly as a theoretical contribution.

I am unfortunately unable to judge the strength of the contribution of this work as the submission is far from my area of research.

**Weaknesses:**

The experimental part of the paper focuses on purely synthetic tasks. While I understand that, as the authors mention, they believe there aren't direct baselines or datasets for this, I still see this as a potential weakness.

A way to greatly strengthen the work would be to have experiments on domains mentioned in the introduction (i.e. metabolic networks).

**Questions:**

Would the authors be able to explain why they think there are no direct baselines? I am not familiar with the surrounding literature, but the problem of estimating a full matrix from a sub-matrix should be a well-studied area. It might be useful to include baselines that do this, even if they are not operating in the transfer learning regime.

Would the authors be able to provide experiments on a real-world task -- or if not provide a justification?

**Limitations:**

None that I am aware.

---

> ### Author Rebuttal · Authors · 2024-08-05
>
> We sincerely thank the reviewer for highlighting the quality of our writing and the value of the problem we tackle. We address their comments in detail below.
>
> ### **We test our algorithms on two real-world transfer tasks.**
>
> In Section 4, lines 271-287, we test our algorithms on two real-world transfer tasks, using the BiGG dataset from the biological networks literature (King et al. 2016) and the Email-EU dataset from the social networks literature (Leskovec and Krevl 2014).
>
> ### **We perform experiments on the domains mentioned in the introduction (metabolic network estimation).**
>
> In Section 4, lines 273-279, we use our algorithms to estimate the metabolic network of _Pseudomonas putida_, a gram-negative bacterium that is studied for its applications to industrial biocatalysis [1] and bioremediation [2]. The full metabolic network for _Pseudomonas putida_ is not known [3]. We use our transfer learning algorithms to estimate its metabolic network for different choices of source organism (Figure 2, left). For a good choice of source organism, our Algorithm 1 achieves mean-squared error comparable to the Oracle with flip probability $p=0.1$.
>
> ### **Comparisons against a new baseline adapted from Levin et al. (JMLR 2022).**
>
> In the global rebuttal, we implement a new baseline transfer method from the literature to compare against our algorithms. See the global rebuttal and attached PDF for results. At a high level, our algorithm is better in most parameter regimes that we test, which we will discuss more below.
>
> We introduce a new model of transfer learning on networks (lines 69-71). To our knowledge, there are no published algorithms with provable guarantees that can be directly applied to our setting. Two key differences of our setting compared to existing works are:
> * _No node data for most nodes in the target network $Q$_. Most works on transfer learning for graphs assume access to some edge data for every node in the target graph (Wang et al. 2018, Levin et al. 2022). Similarly, matrix completion papers typically assume that at least one entry from each row/column of the target matrix is observed (Chatterjee 2015, Simchowitz et al. 2023 and references therein), or otherwise assume low rank/nuclear norm (see [2] and references therein). In our setting, we only observe an $n_Q \times n_Q$ subgraph for $n_Q \ll n$. Hence for most target nodes we observe none of their edges. This is comparable to the MNAR (Missing Not at Random) model in matrix completion [3].
>
> * _No node labels in any network_. We consider a latent variable model in which the relevant features of nodes are not observed. This excludes approaches that rely on observed node labels or features (Tang et al. 2016, Zou et al. 2021, Qiao et al. 2023, Wu et al. 2024).
>
> Nevertheless, for the sake of comparison we implement a new baseline based on a modification of Levin et al. [1]. Specifically, we modify the estimator from their Section 3.3. Their method assumes that full edge data from both P and Q are observed, and that they have the same expectation ($P = Q$). Since this is not true for us, we instead compute the modified MLE:
>
> $$\tilde Q_{ij} = \begin{cases}
> \frac{w_P}{w_P + w_Q} A_{P;ij} + \frac{w_Q}{w_P + w_Q} A_{Q;ij} & i, j \in S \\\\
> A_{P;ij} & \\text{otherwise}
> \end{cases}$$
>
> Where the weights $w_P, w_Q$ are computed as in their paper, based on estimated sub-gamma parameters of the noise for $A_P, A_Q$. This is the only modification we make to their algorithm.
>
> Note that Levin et al. only give theoretical guarantees on the spectral norm $\|| \hat Q - Q \||_2$ of their estimator $\hat Q$. Analyzing the stronger metric of mean-squared error $\frac{1}{n^2} \|| \hat Q - Q \||_F^2$ would require different techniques than their paper.
>
> ### **Matrix completion without transfer should perform poorly in our setting.**
>
> In our setting, we only observe target data on an $n_Q \times n_Q$ submatrix of the adjacency matrix, for $n_Q \ll n$. To apply a matrix completion algorithm, we can certainly zero-pad the target data to obtain some $A_Q \in \\{0,1\\}^{n \times n}$, and apply a standard matrix completion algorithm to this $A_Q$. However, in this case the matrix $A_Q$ would have mostly all-zeroes rows and columns, corresponding to nodes $i \not \in S$. Up to permutation, the zero-padded input matrix would have block structure:
>
> $$A_Q = \begin{bmatrix} A_Q[S, S] & 0 \\\\ 0 & 0 \end{bmatrix}$$
>
> Where $A_Q[S, S] \in \\{0,1\\}^{n_Q \times n_Q}$ contains the observed target data. Note that $n_Q \ll n$ so the matrix is almost all zeroes.
>
> Therefore, any left/right singular vector of $A_Q$ will be of the form $v = \begin{bmatrix} v_S \\\\ 0 \end{bmatrix}$, where $v_S \in \mathbb{R}^{n_Q}$ is a singular vector of $A_S[S,S]$. The singular vectors will contain no information outside of $S$, and therefore the matrix completion algorithm will do poorly.
>
> [1] Nikel, Pablo I., and Víctor de Lorenzo. "Pseudomonas putida as a functional chassis for industrial biocatalysis: from native biochemistry to trans-metabolism." Metabolic engineering 50 (2018): 142-155.
>
> [2] Ward, Patrick G., et al. "A two step chemo-biotechnological conversion of polystyrene to a biodegradable thermoplastic." Environmental science & technology 40.7 (2006): 2433-2437.
>
> [3] Yuan, Qianqian, et al. "Pathway-consensus approach to metabolic network reconstruction for Pseudomonas putida KT2440 by systematic comparison of published models." PloS one 12.1 (2017): e0169437.

---

> > ### Comment · Reviewer_VhmF · 2024-08-08
> >
> > I thank the authors for the detailed response. After having read the discussions with the other reviewers, I would like to increase my score to a 6.

---

> > > ### Author Response · Authors · 2024-08-08
> > >
> > > Dear reviewer VhmF,
> > >
> > > Thank you very much for taking our rebuttal into account and for raising your score.

---

### Official Review · Reviewer_ajJ3 · 2024-07-14

**Soundness:** 3
**Presentation:** 3
**Contribution:** 3
**Rating:** 7
**Confidence:** 3

**Summary:**

This paper investigates transfer learning in the context of estimating latent variable network models.  Specifically, the goal is to estimate the edge probability matrix $Q$ of the target graph using (1) edge data from a source graph $P$ given by its adjacency matrix, and (2) edge data from a vanishingly small subgraph of $Q$ that consists of an $o(1)$ fraction of the nodes in $Q$. The authors propose a transfer learning algorithm (Algorithm 1), which is based on matching quantiles, and demonstrate that it is possible to accurately estimate $Q$ if $P$ and $Q have similar quantile/ranking profiles (per Definition 1.3) and smooth latent variable representations (Assumption 2.1).  Furthermore, in Section 3, the authors focus on Stochastic Block Models, stating the minimax rate under the setup and providing an estimation algorithm (Algorithm 2) that achieves this rate (Proposition 3.4).  The proposed algorithms are then tested with numerical experiments on synthetic and real-world datasets.

**Strengths:**

This work addresses the significant problem of learning graphs/networks from a small subset of edge data by leveraging knowledge from a source graph. This problem is fundamental and has numerous potential practical applications, such as in biological network estimation and social sciences. The paper considers a simple yet expressive mathematical model and presents key findings and supporting arguments clearly. Although primarily theoretical, the results are complemented by numerical experiments that substantiate the potential of the proposed approach.

**Weaknesses:**

While this manuscript makes substantial contributions, there are some areas for improvement to enhance its impact:

**1. Model Assumptions:** The models and assumptions in this work —- latent variable models, H\”older smoothness of the latent functions, and the ranking assumptions between source $P$ and target $Q$ —- are quite standard in theoretical literature. Nevertheless, including in-depth discussions and ablation studies to examine their relevance and applicability in real-world scenarios would help motivate and convince practically oriented readers.

**2. Further Numerical Experiments:** Additional numerical experiments to verify and examine the theoretical results (Theorem 2.2 and Proposition 3.4) would be valuable. This would help validate whether the expected dependence on parameters such as $d, \beta, n_Q$ are sharp or not. Additionally, comparing the performance of the proposed transfer learning algorithms against a true oracle with direct access to the full edge data from $Q$ (which would correspond to what the authors call “oracle” with $p_{flip} = 0$) would properly quantify the cost of transfer and evaluate the effectiveness of the proposed algorithms.  It would also be beneficial to compare the proposed algorithms to other existing algorithms in the literature to highlight the claimed advantages of the proposals.

**Questions:**

1. Based on the description of the Oracle algorithm in lines 246-252, it seems that its performance should be independent of $n_Q$. However, in Figure 2, the Oracle algorithm's performance appears to improve as $n_Q$ ​increases in the first two plots. Could the authors clarify if there is something I am missing?

2. For the Email networks, Algorithm 1 seems to outperform the Oracle (with $p=0.01$ and $p=0.05$). Can the authors provide any insights on how a transfer learning algorithm (i.e., Algorithm 1) can outperform an Oracle algorithm that has full access to the edge data from $Q$?

*Minor suggestions/typos:*
- Line 71:  The order of $|S|$ and $n_Q$ should be swapped, i.e., $n_Q := |S| = o(n)$.
- Lines 102 - 104: The quantifier for $i$ seems missing, e.g., "... *for $i$,* and for all $j \neq i$ ..."
- Line 111: Something should be off here.
- Lines 183 - 184: In the second sentence of Theorem 2.2, it is unclear which parameter corresponds to which function.  I would suggest the authors write for example "Let $f_P$ be $\alpha$-H\"older-smooth and $f_Q$ be $\beta$-H\"older-smooth for $\beta \geq \alpha > 0$, ..."
- Lines 238 - 239: It seems a line break is needed before "(1) Algorithm 2."
- Lines 244 - 245: This line break should be removed.
- Section 4: Please use $p_{flip}$ instead of $p$ for consistency (or vice versa).

**Limitations:**

This is primarily a theoretical work, and the authors discussed the potential limitations of the work and potential future research directions.

---

> ### Author Rebuttal · Authors · 2024-08-05
>
> We sincerely thank the reviewer for highlighting the fundamental nature of our problem, and its numerous potential practical applications. We will fix all typographical errors in the revision.
>
> We address their feedback in detail below.
>
> ### **Relevance and applicability of latent variable models in real-world scenarios**.
>
> Latent variable models are widely used in applied fields such as neuroscience [1], ecology [2], international relations [3], political pscyhology [4], and education research [5]. To help motivate practically oriented readers, we will include more in-depth discussion of these applications in the revision.
>
> ### **Relevance and applicability of the rankings assumption for biological networks.**
>
> Previous works require some form of similarity between networks to enable transfer (Sen et al. 2018, Fan et al. 2019, Baranwal et al. 2020). For example, Kshirsagar et al. 2013 require a _commonality hypothesis_: if pathogens A, B target the same neighborhoods in a protein interaction network, one can transfer from A to B. Our rankings assumption similarly posits that to transfer knowledge from A to B, A and B have similar 2-hop neighborhood structures.
>
> ### **New ablation experiments in the global rebuttal validate our algorithms' error rates.**
>
> We test dependence of Algorithm 1 on $n_Q, d, \beta$ and Algorithm 2 on $n_Q, k_Q$. See the global rebuttal and the attached PDF.
>
> * Experiments include comparisons to a true oracle with access to full edge data from $Q$. See also the discussion in the next heading.
>
> * Experiments include new comparisons to a proposed algorithm in the literature by Levin et al (JMLR 2022). See global rebuttal for a full description.
>
> * Performance of Algorithm 1 and Algorithm 2 both match the trend lines given by theoretical upper bounds (Theorem 2.2 and Proposition 3.4 respectively). Note that we plot theoretical upper bounds assuming each one has a constant of $1.0$, but these  are likely to be larger.
>
> ### **Oracle with $p = 0.0$ outperforms transfer algorithms in our new experiments.**
>
> An oracle with $p = 0.0$ corresponds to a non-transfer setting where all edges from the target graph $Q$ are observed. Note that in this case, the value of $n_Q$ does not matter because edges incident to nodes outside of $S$ never get flipped. In lines 251-252, we noted that when $p=0.0$, the Oracle error for $\beta$-smooth graphons on $d$-dimensional latent variables will be $O(n^{-\frac{2 \beta}{2\beta + d}})$ (Xu 2018), which is less than the error bound of Theorem 2.2.
>
> In our new experiments we implement this oracle and indeed find that it outperforms our transfer algorithms in the regimes where its theoretical upper bound is better than our theoretical upper bounds. In particular, our upper bounds scale with $n_Q$, rather than $n$, so for $n_Q \ll n$ transfer should generally be worse. Note that for our Algorithm 1 and for this version of the oracle, lower bounds are unknown (see our discussion in lines 193-196). Still, these experimental results should help quantify the cost of transfer and evaluate the effectiveness of our proposed algorithms.
>
> ### **The Oracle depends on $n_Q$ if $p > 0$.**
>
> Following the equation below line 249, the oracle observes the unbiased $Y_{ij} \sim Bernoulli(Q(x_i, x_j))$ if $i, j \in S$ are both part of the set of observed target nodes $S$ such that $\lvert S \rvert = n_Q$. If $i \not \in S$ or $j \not \in S$ then the oracle observes a possibly flipped version of $Y_{ij}$ when $p > 0$. Therefore as $n_Q$ grows, the oracle should improve, because none of the nodes in $S$ can be flipped.
>
> We will clarify this point in the revision.
>
> ### **The Oracle performs much better on the Email-EU task with $p = 0.0$ than with $p=0.01$.**
>
> Rerunning our Email-EU experiment with a flip probability of $p=0.0$, and comparing to our existing results from Figure 2, we have the following. We use ($\dagger$) to denote a new experiment.
>
> | $n_Q$  | Method | Target  | MSE (Median, 50 Trials)    |
> ---- | ---- | ---- | ---- |
> | 20  | Oracle, $p=0.0$ ($\dagger$)    | Email-EU Days 81-160      | 0.003797 |
> | 20  | Oracle, $p=0.01$    | Email-EU Days 81-160      | 0.007312 |
> | 20  | Our Alg. 1 | Email-EU Days 81-160      | 0.007240 |
> | 20  | Oracle, $p=0.0$ ($\dagger$)  | Email-EU Days 561-640      | 0.004100 |
> | 20  | Oracle, $p=0.01$    | Email-EU Days 561-640      | 0.007620 |
> | 20  | Our Alg. 1 | Email-EU Days 561-640      | 0.007591 |
>
> For this dataset, a flip probability of $p = 0.0$ versus $p = 0.01$ makes a substantial difference. Note that all MSE values on the right-hand side of Figure 2 are within $[0.007, 0.008]$, whereas the Oracle with $p=0.0$ has MSE $\approx 0.04$. Note that the Oracle with $p=0.0$ does not depend on $n_Q$ since it access the full, unbiased edge data from $Q$.
>
> We believe this difference is because the email networks in Figure 2 are quite sparse, with median degree $\leq 0.007n$ (see Table 2). Therefore even introducing a $0.01$ probability of edge flips makes the Oracle substantially worse on these networks.
>
> [1] Ren, Mingyang, et al. "Consistent estimation of the number of communities via regularized network embedding." Biometrics 79.3 (2023): 2404-2416.
>
> [2] Trifonova, Neda, et al. "Spatio-temporal Bayesian network models with latent variables for revealing trophic dynamics and functional networks in fisheries ecology." Ecological Informatics 30 (2015): 142-158.
>
> [3] Cao, Xun, and Michael D. Ward. "Do democracies attract portfolio investment? Transnational portfolio investments modeled as dynamic network." International Interactions 40.2 (2014): 216-245.
>
> [4] Barberá, Pablo, et al. "Tweeting from left to right: Is online political communication more than an echo chamber?." Psychological science 26.10 (2015): 1531-1542.
>
> [5] Sweet, Tracy M., et al. "Hierarchical network models for education research: Hierarchical latent space models." Journal of Educational and Behavioral Statistics 38.3 (2013): 295-318.

---

> > ### Comment · Reviewer_ajJ3 · 2024-08-10
> > **Response to the Authors' Rebuttal**
> >
> > I thank the authors for addressing my questions and concerns.  I trust that the authors will incorporate the new experiment and additional explanations into their revision.  With that understanding, I am inclined to raise my rating to a 7.

---

> > > ### Author Response · Authors · 2024-08-10
> > >
> > > Dear reviewer ajJ3,
> > > Thank you very much for taking our rebuttal into account and for raising your score. We will incorporate the new experiments and additional explanations into the revision.

---

### Author Rebuttal · Authors · 2024-08-05

We sincerely thank all the reviewers for their encouraging comments and for acknowledging the significance of our work. We will fix all typographical errors in the revision.

In this global rebuttal, we will mainly discuss the new experiments, attached as PDF.

### **New experiments quantify the dependence of Algorithms 1 and 2 on all parameters [Reviewer ajJ3].**

Each experiment tests the dependence of one of our algorithms on a particular parameter (see the attached PDF).

Algorithm 1 is for general latent variable models. The error (Theorem 2.2) depends on the smoothness $\beta$ of the target graph, the number of observed target nodes $n_Q$, and the dimension of the latent variables $d$.

Algorithm 2 is for Stochastic Block Models. The error (Proposition 3.4) depends on the number of communities $k_Q$ in the target graph, and the number of observed target nodes $n_Q$. Note that Proposition 3.4 also depends logarithmically on the minimum community size of $Q$, but this is less significant.

### **Comparisons against a new baseline adapted from Levin et al. (JMLR 2022) [Reviewers ajJ3 and VhmF].**

We introduce a new model of transfer learning on networks (lines 69-71). To our knowledge, there are no published algorithms with provable guarantees that can be directly applied to our setting. Two key differences of our setting with existing works are:
* _No node data for most nodes in the target network $Q$_. Most works on transfer learning for graphs assume access to some edge data for every node in the target graph (Wang et al. 2018, Levin et al. 2022). Similarly, matrix completion papers typically assume that at least one entry from each row/column of the target matrix is observed (Chatterjee 2015, Simchowitz et al. 2023 and references therein), or otherwise assume low rank/nuclear norm (see [1] and references therein). In our setting, we only observe an $n_Q \times n_Q$ subgraph for $n_Q \ll n$. Hence for most target nodes we observe none of their edges. This is comparable to the MNAR (Missing Not at Random) model in matrix completion [2].

* _No node labels in any network_. We consider a latent variable model in which the relevant features of nodes are not observed. This excludes approaches that rely on observed node labels or features (Tang et al. 2016, Zou et al. 2021, Qiao et al. 2023, Wu et al. 2024).

Nevertheless, we implement a new baseline based on a modification of the estimator in Section 3.3 of Levin et al., 2021. They assume that full edge data from both P and Q are observed, and $P=Q$. Since this is not true for us, we instead compute the modified MLE:

$$\tilde Q_{ij} = \begin{cases}
\frac{w_P}{w_P + w_Q} A_{P;ij} + \frac{w_Q}{w_P + w_Q} A_{Q;ij} & i, j \in S \\\\
A_{P;ij} & \\text{otherwise}
\end{cases}$$

Where $w_P, w_Q$ are computed as in their paper, based on estimated sub-gamma parameters of the noise for $A_P, A_Q$. Akin to their adjacency spectral embedding, which assumes known rank of $Q$, we use Universal Singular Value Thresholding to obtain $\hat Q$ from $\tilde Q$.

Note that while we can plot theoretical guarantees for the mean squared error $\frac{1}{n^2} \|| \hat Q - Q \||_F^2$ of both our algorithms' $\hat Q$ and the oracle's $\hat Q$, Levin et al. only give theoretical guarantees on the spectral norm $\|| \hat Q - Q \||_2$ for their estimator $\hat Q$. Analyzing the stronger metric of mean-squared error would require different techniques than their paper.

### **Comparisons against the Oracle with $p = 0.0$ [Reviewer ajJ3].**

As suggested by reviewer ajJ3, we compare against the Oracle with flip probability $p = 0.0$. This corresponds to a non-transfer setting in which full data from the target graph $Q$ are observed. In this case, the value of $n_Q$ does not matter since edges incident to nodes outside of $S$ never get flipped. In lines 251-252, we note that such an oracle should be better than Algorithm 1 for smooth graphons. This is supported by our experiments.

### **Takeaways from our new experiments (see attached PDF) [Reviewers ajJ3 and VhmF].**

Our main takeaway is that the performance of Algorithms 1 and 2 matches the trend of our theoretical upper bounds (modulo constants). These validate our analyses and show the applicability of our methods in a wide variety of regimes, complementing our existing experiments on real-world data.

By comparing against the Oracle with full edge data from $Q$, we help quantify the cost of transfer. As we would expect, the Oracle with $p = 0$ consistently attains lower mean-squared error than our Algorithms. This is to be expected because our transfer algorithms do not access the full target data from $Q$.

The baseline adapted from Levin et al. is worse than our Algorithms. This is not surprising because it is not designed for a setting in which $P \neq Q$ and a vanishing fraction of $Q$ is observed.

### **Clarifying the effect of sparsity on the choice of graph distance [Reviewers BaBd and TFsn].**

We use the common neighbors graph distance (Definition 1.2) to capture local graph structure in Algorithm 1. Other graph distances have been used in the literature as well (Zhang et al. 2017, Mukherjee and Chakrabarti 2019, Mao et al. 2021), but ours is useful for technical reasons (see response to Reviewer BaBd).

For expected degree $\tilde o(n^{1/2})$, our graph distance (Definition 1.2) fails to concentrate (Mao et al. 2021). However, the same authors show that a modified common neighbors concentrates for expected degree $\tilde \Omega(n^{1/3})$. This suggests that variations on the graph distance might ensure our Algorithm 1 works for sparser graphs.

[1] Xiang, Yunhua, et al. "On the optimality of nuclear-norm-based matrix completion for problems with smooth non-linear structure." JMLR 2023.

[2] Ma, Wei, and George H. Chen. "Missing not at random in matrix completion: The effectiveness of estimating missingness probabilities under a low nuclear norm assumption." NeurIPS 2019.

---

### Decision · Program_Chairs · 2024-09-25

**Decision:**

Accept (poster)

**Comment:**

There is consensus that this submission should be accepted.